# Thermodynamics of quantum-jump trajectories of open quantum systems subject to stochastic resetting

Gabriele Perfetto[1]$\star$, Federico Carollo[1] and Igor Lesanovsky[1,2]

**1** Institut für Theoretische Physik, Eberhard Karls Universität Tübingen, Auf der Morgenstelle 14, 72076 Tübingen, Germany.
**2** School of Physics and Astronomy and Centre for the Mathematics and Theoretical Physics of Quantum Non-Equilibrium Systems, The University of Nottingham, Nottingham, NG7 2RD, United Kingdom.

$\star$ gabriele.perfetto@uni-tuebingen.de

October 5, 2022

## Abstract

We consider Markovian open quantum systems subject to stochastic resetting, which means that the dissipative time evolution is reset at randomly distributed times to the initial state. We show that the ensuing dynamics is non-Markovian and has the form of a generalized Lindblad equation. Interestingly, the statistics of quantum-jumps can be exactly derived. This is achieved by combining techniques from the thermodynamics of quantum-jump trajectories with the renewal structure of the resetting dynamics. We consider as an application of our analysis a driven two-level and an intermittent three-level system. Our findings show that stochastic resetting may be exploited as a tool to tailor the statistics of the quantum-jump trajectories and the dynamical phases of open quantum systems.

# 1   Introduction

An open quantum system is a system which is in interaction with its surrounding environment, see, e.g., Refs. [1–3]. Within the Markovian approximation of weak coupling between the system and the environment, the ensuing system dynamics is governed by a master equation through the so-called Lindblad generator [4, 5]. This dynamics describes the non-unitary time-evolution of the state of the quantum system averaged over all the *quantum trajectories*, i.e., the time-evolution of the quantum state conditioned on the individual realisations of the system-environment interaction [6–8]. In an experiment, the interaction between system and environment can indeed be monitored through a detector (see Fig. 1), performing a continuous measurement of the exchange of quanta between them. In the case of atomic decay, for instance, the time-record of this measurement corresponds to the detection of a sequence of photons emitted by the system into the environment. To each time-record of the measurement outcomes, which we call here *quantum-jump trajectory*, there is an associated quantum trajectory.

In the framework of Markovian open quantum systems, the statistical properties of the quantum-jump trajectories can be unveiled through a "thermodynamic" formalism, first proposed in Ref. [9] and then applied in Refs. [10–20], with Ref. [21] providing a review on the subject. This approach, which was originally developed for classical systems [22–29], considers ensembles of quantum-jump trajectories in the same way statistical mechanics deals with ensembles of configurations. A paradigmatic order parameter classifying the quantum-jump trajectories is the activity, i.e., the total number of detected quantum jumps up to a certain observation time $t$. The activity is, however, an order parameter of intrinsic *dynamical nature* since it depends on the underlying dynamical trajectories. This is in contrast to the case of equilibrium order parameters, e.g., the magnetization, which depends on static equilibrium configurations. The rare (or atypical) fluctuations of the activity for a long observation time $t$ can be characterized using the large deviation theory, see, e.g., Refs. [30–32]. The large deviation function, in particular, quantifies the asymptotic concentration of the probability distribution of the activity around the average value for long times $t$.

The large deviation function can be computed from the corresponding scaled cumulant generating function $\theta(k)$ (SCGF) [30–32], with $k$ the variable conjugate to the activity. The

SCGF generalizes the concept of free energy to the dynamical realm of quantum-jump trajectories. In Markovian open quantum systems there exists a general recipe to compute $\theta(k)$. This approach, which is widely known in classical stochastic systems from Refs. [33, 34], consists of tilting the probability measure of the trajectories by inserting a Gibbs-like weighting factor, that depends on the activity of a given trajectory. The concomitant change of the trajectory probability measure translates into a change of the Lindblad dynamical generator, that governs the time evolution of the system. The ensuing generator – usually named tilted Lindblad generator [9, 13, 17, 19] – depends on the counting variable $k$. It, however, does not generate a physical, probability-preserving, dynamics for $k \neq 0$. Note, that it has been shown in Ref. [9, 17] that a probability-preserving and $k$ dependent Lindblad generator can be constructed starting from the tilted generator via the so-called quantum Doob transform. The Doob transformed generator displays as typical quantum-jump trajectories the rare trajectories of the original Lindblad generator. The Doob transformed generator can be consequently used to efficiently simulate rare trajectories of a Markovian-Lindblad quantum dynamics. The SCGF can be instead computed as the eigenvalue of the tilted generator with the largest real part. Such tilting can actually also be used to study the large deviations of time-integrated currents in non-equilibrium unitary systems, as shown in Refs. [35–38]. Since in this unitary case the dynamics is deterministic and the initial state is fluctuating, the tilting is in this case applied to the initial probability measure over configurations.

There is currently a growing interest in the non-equilibrium physics community, that concerns dynamical processes with *stochastic resetting*. The latter has been first proposed to model stochastic processes, such as animal foraging for food [39, 40] or a person searching for an object, where the searchers return at random times after an unsuccessful period of search to the starting position, where the food was first successfully found or the object was seen the last time. Another example of a similar stochastic process is provided by biophysics models for proteins searching for a binding site on DNA [41–43]. The common aspect of these processes is that they all fall within the class of "intermittent search strategies" [44]. These search strategies are made by combining phases of slow motion moves, where the searcher attempts to detect the target, and long-range relocation moves at stochastic times, where the searcher is not directly looking for the target. In this framework, the presence of long-range moves at random times, the resets, has been shown to lead to efficient searching algorithms since the mean first passage time can be minimized as a function of the resetting parameters/distribution [45–52]. From the physics point of view, stochastic resetting has been largely investigated as a paradigmatic protocol to generate non-equilibrium stationary states, since resets hinders the system relaxation to an equilibrium state by repeatedly bringing the system back to its initial state [46, 53–55]. Furthermore, stochastic resetting can lead to dynamical phase transitions in the relaxation towards the non-equilibrium stationary state [56].

The simplest, yet instructive and not trivial, case where the aforementioned features of the resetting dynamics can be observed is the one proposed in Refs. [46, 53] of a classical diffusing particle whose position is reset to its initial value at constant rate $\gamma$, i.e., Poissonian resetting. The more general case, where the times between consecutive resets are distributed according to a generic waiting time distribution $p(\tau)$, can be analyzed within the so-called *renewal equation* approach, which relates the dynamics in the presence of resetting to the one where resets are absent, as shown in Refs. [57–65]. At the level of the master equation, non-Poissonian resetting is typically harder to treat than the Poissonian case. The reason is that in the former case the resetting rate is time dependent and the master equation is consequently non-

Markovian, as proved in Ref. [55]. The renewal approach is effective also for the computation of the large deviation statistics of dynamical, i.e., time-integrated, observables of the resetting dynamics [66–69]. In Refs. [66, 67], in particular, a renewal equation relating the moment generating function of the reset process to the one without resetting has been derived in the case of Poissonian resetting.

In the quantum realm, stochastic resetting amounts to a re-initialization of the quantum state of the system to a chosen reset state at randomly distributed times. The Lindblad equation governing the evolution of a Markovian open quantum system undergoing Poissonian resetting at rate $\gamma$ has been first formulated in Refs. [70, 71] and subsequently analyzed in Refs. [19, 72–74]. Within these works, stochastic resetting is considered as a further Markovian dissipative process which is added to the ones already present in the Lindblad equation dictating the open system dynamics between consecutive resets. The latter are therefore quantum jumps that project the state of the system to the reset state at a constant rate $\gamma$, independent from the state of the system. Closed quantum systems subject to Poissonian resetting have been, instead, analyzed in Ref. [75] exploiting the renewal equation approach. In Ref. [76], both the renewal equation and the master equation approach have been pursued for an arbitrary resetting distribution $p(\tau)$. Here, the resulting evolution equation for the density matrix, assumes the form of a generalized Lindblad equation. The main difference with the Lindblad equation, which is recovered in the case of Poissonian resetting, lies in the fact that the generalized Lindblad equation in its more natural form involves an integration over time and it is therefore non-Markovian. These recent works show how non-Poissonian resetting can be incorporated in the framework of a generalized dissipative dynamics. However, it is not clear how the thermodynamics of trajectories formalism of Ref. [9] can be formulated and applied to non-Poissonian resetting.

For such case, indeed, there are no analytical methods to obtain the SCGF as it cannot be computed as the largest real eigenvalue of the tilted generator since the latter is time dependent. At the numerical level, the calculation of the SCGF, which is equivalent to the calculation of the far tails of the activity probability distribution at long times, is as well extremely demanding. Rare events are, indeed, exponentially suppressed as the time $t$ increases because of the large deviation principle, and the sampling of the associated probability distribution tails becomes increasingly inefficient. The aforementioned quantum Doob transform of Ref. [9, 17], moreover, does not apply here because of the non-Markovian character of the non-Poissonian resetting. As a consequence, there is, at present to the best of our knowledge, no efficient method to simulate rare quantum-jump trajectories for non-Poissonian resetting superimposed to an open-Lindblad quantum dynamics. The availability in the latter context of analytical results for the SCGF is, therefore, highly desirable and relevant.

In this manuscript, we fill this gap by calculating the exact large deviation function of the dynamical activity and the associated SCGF $\theta(k)$ for Markovian open quantum systems subject to non-Poissonian resetting. Our derivations are valid for arbitrary waiting time distribution $p(\tau)$. In terms of formulas, the main result of the manuscript for the calculation of the SCGF $\theta(k)$ is given by Eqs. (50)-(55). To our knowledge, this is the first exact evaluation of the large deviation statistics of the activity for dissipative quantum systems subject to non-Poissonian resetting, i.e., for a non-Markovian dynamical generator. Note, that the large deviation statistics of the activity for a non-Markovian dynamics has previously been exactly computed in Ref. [16] for a different protocol, where a control-feedback mechanism applied to single quantum trajectories — spoiling the Markovian character of the dynamics — is superimposed to a Lindbladian master equation evolution.

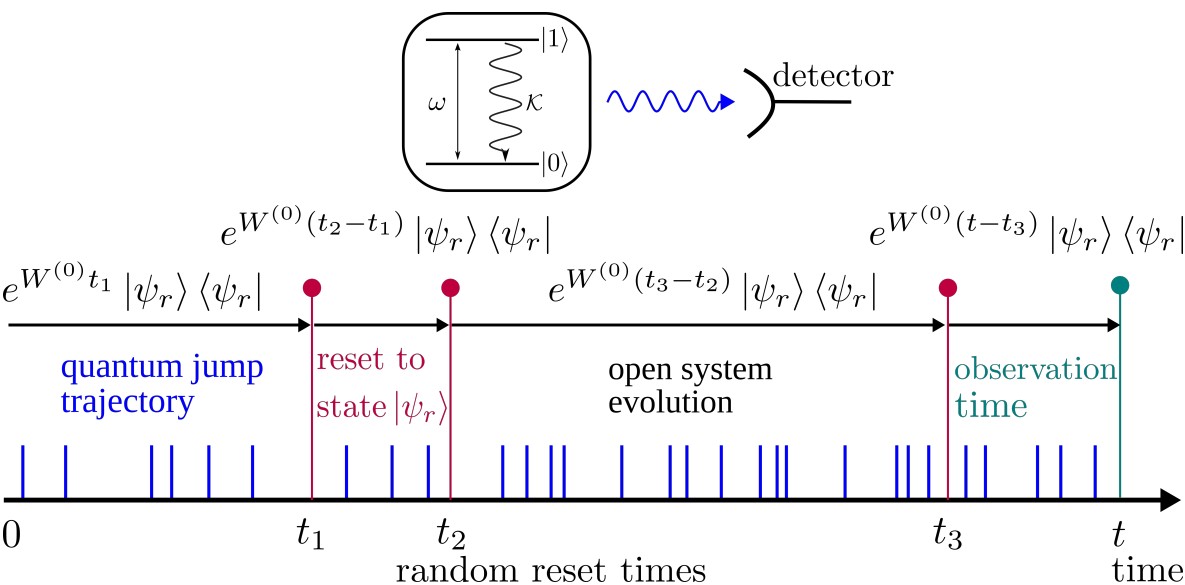

Figure 1: **Open quantum system with resets.** Schematic representation of a quantum-jump trajectory over an observation time interval $(0, t)$ in an open quantum system subject to stochastic resetting. In the Figure, a two-level system with Rabi frequency $\omega$ and decay rate $\mathcal{K}$ is depicted as an example. The system is initialized in the reset state $|\psi_r\rangle$ and it is then reset to $|\psi_r\rangle$ at randomly distributed times, represented by the purple vertical lines. In the Figure, the number $R$ of resets prior to the observation time $t$ is equal, as an example, to 3. Between consecutive resets the system evolves according to a Markovian dissipative dynamics with Lindbladian generator $W^{(0)}$. The signal associated with the stochastic quantum jumps of $W^{(0)}$ — in the case of the two-level system the emission of a photon — is recorded by a detector and it is represented by the blue vertical lines. Quantum-jump trajectories in the presence of resetting can be accordingly split into $R$ independent trajectories, each starting from $|\psi_r\rangle$. In the manuscript, we exactly compute the large deviation statistics of the total number of quantum jumps over a long observation time $t$.

Our results are obtained by deriving the non-Markovian tilted generator with two equivalent approaches. First, we do so by conditioning, as shown pictorially in Fig. 1, the quantum-jump trajectories of the system on the number $R$ of resets that happened prior to the observation time $t$. This yields a tilted non-Markovian generator from which we derive a renewal equation relating the moment generating function in the presence of resets to the one of the purely Markovian open dynamics. This extends the results of Refs. [66, 67] to the more complex case of non-Poissonian resetting in dissipative quantum systems. Second, we derive the tilted non-Markovian generator directly by formulating a renewal equation connecting the tilted density matrix with resets, $\rho(k, t)$ to the density matrix without resets, $\rho^{(0)}(k, t)$. We eventually particularize our results to some simple, yet relevant, quantum-optical systems. Namely, we consider a driven two-level atom subject to incoherent decay, as in Fig. 1, and a three-level system, which can show intermittency [9, 12, 16, 77] in the recorded emissions. In both the cases we establish stochastic resetting as a valuable tool to analyze and tailor the dynamical activity of the system depending on the resetting time scale and on the chosen reset state. We emphasize that the ability to exactly calculate the SCGF is remarkable even for apparently simple quantum optics systems such as driven two and three level atoms. On

the basis of the aforementioned discussion, indeed, the SCGF for non-Poissonian resetting cannot be analytically evaluated via other means.

The manuscript is organized as follows. In Sec. 2, we review the main concepts regarding the thermodynamics of quantum-jump trajectories in Markovian open quantum systems and stochastic resetting. In Sec. 3, we present the main result of the manuscript, which is the exact calculation of the large deviation statistics for open quantum systems subject to non-Poissonian resetting. In Sec. 4, we apply the general results of Sec. 3 to two and three-level atomic systems. In Sec. 5 we draw our conclusions. More technical aspects are detailed in the Appendices A and B.

## 2    Introduction to quantum-jump trajectories and stochastic resetting

In this Section we review the main concepts underlying large deviation theory in Markovian open quantum systems, described by the Lindblad equation, in combination with stochastic resetting. In Subsec. 2.1 we review the thermodynamics of quantum-jump trajectories formalism for Markovian open quantum systems. We will also discuss how the large deviation function associated with the probability distribution of the number of quantum jumps in a given time interval can be computed by using a suitably tilted generator. In Subsec. 2.2 we consider the introduction of stochastic resetting on top of a Lindblad dynamics. In particular, we show that for generic resetting distribution the ensuing dynamics is non-Markovian and that it is described by a generalized Lindblad equation. In Subsec. 2.3 we particularize the analysis to resetting at constant rate $\gamma$ — Poissonian resetting. In the latter case the dynamics in the presence of reset simplifies to a Lindblad form, which allows for the calculation of the quantum-jump statistics as the largest real eigenvalue of the tilted generator.

### 2.1    Thermodynamics of quantum-jump trajectories in Markovian open quantum systems

We consider a quantum system, described by the density matrix $\rho(t)$, which is interacting with its surrounding environment. Under appropriate assumptions concerning the timescales of the dynamics of the environment and the coupling of the latter with the system, the time evolution of $\rho(t)$ is Markovian and it is given by the Lindblad master equation [4,5]

$$\frac{\mathrm{d}}{\mathrm{d}t}\rho(t) = W[\rho(t)] = -i\left(H\rho(t) - \rho(t)H\right) + \sum_{\mu=1}^{N_L}\left(L_\mu\rho(t)L_\mu^\dagger - \frac{1}{2}\{L_\mu^\dagger L_\mu, \rho(t)\}\right). \tag{1}$$

Here, $\mu = 1, 2 \ldots N_L$, where $N_L$ is the number of jump operators $L_\mu$, $H$ is the Hamiltonian of the system, and $\{.,.\}$ denotes the anti-commutator. We shall consider throughout the manuscript the case where the jump operators $L_\mu$ and the Hamiltonian $H$ are time independent. Equation (1) is trace preserving at any time, given that $\frac{\mathrm{d}}{\mathrm{d}t}\mathrm{Tr}[\rho(t)] = 0$. Moreover, the Lindblad master equation is linear in $\rho(t)$. Consequently, the central object controlling the time evolution is the time-independent generator $W$, which is usually called Lindbladian superoperator. We use henceforth the notation with the square brackets , e.g., $W[X]$ in Eq. (1), for the action of a superoperator $W$ on an operator $X$. This must not be confused with a

commutator. It is useful to rewrite Eq. (1) as

$$\frac{\mathrm{d}}{\mathrm{d}t}\rho(t) = -i(H_{\mathrm{eff}}\rho(t) - \rho(t)H_{\mathrm{eff}}^{\dagger}) + \mathcal{L}[\rho(t)] = \mathcal{L}_0[\rho(t)] + \sum_{\mu=1}^{N_L}\mathcal{L}_\mu[\rho(t)] \tag{2}$$

where we have introduced the non-Hermitian effective Hamiltonian

$$H_{\mathrm{eff}} = H - \frac{i}{2}\sum_{\mu=1}^{N_L}L_\mu^{\dagger}L_\mu, \quad \text{with} \quad \mathcal{L}_0[\rho] = -iH_{\mathrm{eff}}\rho + i\rho H_{\mathrm{eff}}^{\dagger}, \tag{3}$$

and defined the superoperators

$$\mathcal{L} = \sum_{\mu=1}^{N_L}\mathcal{L}_\mu, \quad \text{with} \quad \mathcal{L}_\mu[\rho] = L_\mu\rho L_\mu^{\dagger}, \quad \text{for} \quad \mu = 1, 2 \ldots N_L. \tag{4}$$

The formal solution of the Lindblad equation in the form of Eqs. (2)-(4) can be written as

$$\rho(t) = \mathcal{S}(t,0)[\rho(0)] + \sum_{N=1}^{\infty}\int_0^t \mathrm{d}t_N \int_0^{t_N}\mathrm{d}t_{N-1}\ldots\int_0^{t_2}\mathrm{d}t_1 \mathcal{S}(t,t_N)\mathcal{L}\mathcal{S}(t_N,t_{N-1})\ldots\mathcal{L}\mathcal{S}(t_1,0)[\rho(0)], \tag{5}$$

with the initial condition $\rho(0)$ and the superoperator

$$\mathcal{S}(t,t_0)[\rho] = e^{-iH_{\mathrm{eff}}(t-t_0)}\rho\,e^{iH_{\mathrm{eff}}^{\dagger}(t-t_0)}. \tag{6}$$

Equations (5) and (6) can be derived from Eqs. (1)-(4) as a generalized Dyson series. The Dyson series is a perturbative construction of the unitary time evolution operator $U$ of quantum mechanics carried on in the interaction picture. Within this picture, the Hamiltonian is decomposed as $H = H_0 + V$, with $V$ considered as a, possibly time-dependent, perturbation. The unitary time evolution operator is then decomposed as a series $U = \sum_{n=0}^{\infty}U_n$ of terms $U_n$, where $n$ actions of the perturbations are present interspersed with the unperturbed evolution according to $H_0$. In the present case of a dissipative-Lindblad dynamics, the comparison with the Dyson series of quantum mechanics suggests to consider $\mathcal{L}$ as a "perturbation" and $\mathcal{S}(t,t_0)$ as the "unperturbed" evolution. The Lindblad evolution according to the generator $W$ is then obtained by integrating over all the possible realizations of the perturbation $\mathcal{L}$ interspersed with the unperturbed evolution $\mathcal{S}$. The corresponding formulas are Eqs. (5) and (6). The latter representation of the Lindblad generator $W$ has been, to our knowledge, first reported in Ref. [8], in discussing the emission statistics of two and three-level atomic systems. References [6,7,78] provide an extensive discussion of the subject. In Equation (5) the product of superoperators $\mathcal{L}\mathcal{S}$ denotes their composite action on an operator $X$ as $\mathcal{L}\mathcal{S}[X] = \mathcal{L}[\mathcal{S}[X]]$. Note, that $\mathcal{S}(t,t_0) = \mathcal{S}(t-t_0,0)$, since the effective Hamiltonian $H_{\mathrm{eff}}$ is time independent. Equations (5) and (6) have an interesting physical interpretation: the density matrix $\rho(t)$ at time $t$ is the result of the averaging over all the possible *quantum trajectories*, which are obtained upon interspersing the deterministic time evolution governed by the effective Hamiltonian $H_{\mathrm{eff}}$ with $N = 0, 1, \ldots \infty$ jumps taking place at stochastic times $t_1, t_2 \ldots t_N$ [1]. Each jump, which is represented by one of the $L_\mu$ operators in Eq. (2), has a direct effect on the

---

[1]The effective Hamiltonian $H_{\mathrm{eff}}$ does not only control the time evolution between consecutive jumps, but it also determines the times when jumps occur.

environment. The corresponding "emission signal" in the environment can be detected, i.e. counted, and the record of these counts generates a so-called *quantum-jump trajectory* [9]. A typical example in quantum optics corresponds to the emission (absorption) of a photon to (from) the environment. In the following — for the sake of simplicity [cf. the discussion after Eq. (9) below] — we shall consider the case $N_L = 1$, where the set of jump operators contains only one element. Accordingly, we denote as $L_1$ the jump operator whose counting statistics is of interest. We, however, emphasize that the generalization to an arbitrary number $N_L$ of jump operators is straightforward and does not bear additional conceptual difficulties.

Quantum-jump trajectories can be classified by taking the number $N$ of recorded jumps associated with $L_1$ up to time $t$ as a dynamical order parameter. This observable, which as we already mentioned in the introduction is often called activity, is a random variable because of the stochastic nature of the quantum-jump trajectories. It is accordingly described by the probability $P_t(N)$ of observing $N$ jumps in the interval $(0, t)$, which can be written as [8]

$$P_t(N) = \text{Tr}[\rho(N, t)]. \tag{7}$$

Here, $\rho(N, t)$ is the (conditional) density matrix projected onto the subspace of $N$ jumps. The density matrix $\rho(t)$ is obtained upon summing over all the possible values of $N$

$$\rho(t) = \sum_{N=0}^{\infty} \rho(N, t). \tag{8}$$

From Eqs. (5) and (6), with $N_\mu = 1$, one can immediately identify $\rho(N, t)$ as

$$\rho(0, t) = \mathcal{S}(t, 0)[\rho(0)], \text{ for } N = 0, \qquad \rho(N, t) = \mathcal{V}(N, t, 0)[\rho(0)], \text{ for } N = 1, 2, \dots$$

$$\mathcal{V}(N, t, t_0) = \int_{t_0}^{t} dt_N \int_{t_0}^{t_{N-1}} dt_{N-1} \cdots \int_{t_0}^{t_2} dt_1 \mathcal{S}(t, t_N) \mathcal{L}_1 \mathcal{S}(t_N, t_{N-1}) \dots \mathcal{L}_1 \mathcal{S}(t_1, t_0). \tag{9}$$

The superoperator $\mathcal{V}(N, t, t_0)$ in the last line of the previous equation propagates the state at time $t_0$ to time $t$, integrating over all the quantum trajectories with $N$ jumps taking place in the time interval $(t_0, t)$. The generalization of the expression of $\mathcal{V}(N, t, t_0)$ to the case of an arbitrary number $N_L$ of jump operators is straightforward to do, albeit it is more cumbersome to write than Eq. (9), given that one needs to account for quantum trajectories with an arbitrary number of jumps associated with $L_\mu$, with $\mu = 1, 2, 3 \dots N_L$. From Eq. (6), it immediately follows that $\mathcal{V}(N, t, t_0) = \mathcal{V}(N, t - t_0, 0)$. The definition in Eq. (8) also implies the property

$$\sum_{N=0}^{\infty} \mathcal{V}(N, t, t_0) = e^{(t-t_0)W}, \tag{10}$$

where $W$ is the Lindblad superoperator in Eq. (1). Upon differentiating Eq. (9) with respect to time $t$ one has, as shown in Ref. [8], that the conditional density matrices $\rho(N, t)$ satisfy a hierarchy of equations

$$\frac{d}{dt} \rho(N, t) = \mathcal{L}_0[\rho(N, R, t)] + \mathcal{L}_1[\rho(N - 1, t)](1 - \delta_{N,0}) \tag{11}$$

which relates the evolution of $\rho(N, t)$ to that of $\rho(N - 1, t)$. The first term on the right-hand side of the previous equation describes the dynamics between consecutive quantum jumps with

the effective Hamiltonian $H_{\text{eff}}$ in Eq. (3). The second term on the right-hand side represents, instead, the gain term due to quantum trajectories where the $N_{th}$ jump happens at time $t$.

The number $N$ of recorded jumps up to time $t$ is a time-integrated observable. It has been then shown, in Ref. [9] for the Lindblad dynamics, that the probability $P_t(N)$ in Eq. (7) at long times obeys the large deviation principle [30, 31]

$$P_t(N) \asymp \exp(-tI(n)), \quad \text{as} \quad t \to \infty, \tag{12}$$

with $n = N/t$. The symbol $\asymp$ denotes the equality between the right and the left hand side in logarithmic scale as $t \to \infty$. The large deviation principle in Eq. (12) states that the leading asymptotic dependence of $P_t(N)$ is captured by the large deviation or rate function $I(n)$. The latter is a convex, non-negative function which has a zero at the average and most probable value $\langle n \rangle = \langle N \rangle /t$, with $I(n = \langle n \rangle) = 0$. The probability $P_t(N)$ at long times therefore peaks exponentially around the average value. The large deviation function $I(n)$ is conveniently computed from the scaled cumulant generating function $\theta(k)$ (SCGF), which is defined as

$$G(k,t) = \sum_{N=0}^{\infty} P_t(N)e^{-kN} \asymp \exp(t\,\theta(k)), \quad \text{as} \quad t \to \infty. \tag{13}$$

Here, we have introduced the moment generating function (MGF) $G(k,t)$, which has the physical meaning of a dynamical partition function, see, e.g., Refs. [9, 23, 24, 27], with $k \in \mathbb{R}$ the real variable conjugate to the activity. The link between the SCGF $\theta(k)$ and the large deviation function $I(n)$ is provided by the Gärtner-Ellis theorem [30, 31], which states that if $\theta(k)$ exists and is differentiable for $k \in \mathbb{R}$, then the large deviation principle in Eq. (12) is satisfied and $I(n)$ can be computed as

$$I(n) = \sup_{k \in \mathbb{R}} \{-kn - \theta(k)\}. \tag{14}$$

For a Lindblad dynamics the MGF in Eq. (13) can be written as

$$G(k,t) = \sum_{N=0}^{\infty} \text{Tr}[\rho(N,t)]e^{-kN} = \text{Tr}\left[\sum_{N=0}^{\infty} \rho(N,t)e^{-kN}\right] = \text{Tr}[\rho(k,t)]. \tag{15}$$

In the first equality of Eq. (15) we used Eq. (7) and we defined the *tilted*, or biased, density matrix $\rho(k,t)$ as the discrete Laplace transform of $\rho(N,t)$:

$$\rho(k,t) = \sum_{N=0}^{\infty} \rho(N,t)e^{-kN}. \tag{16}$$

Note, that $\rho(k=0,t) = \rho(t)$ from Eq. (8). The calculation of the SCGF $\theta(k)$ is accomplished by decoupling the hierarchy of equations (11) using the Laplace transform in Eq. (16). This yields the equation

$$\frac{\mathrm{d}}{\mathrm{d}t}\rho(k,t) = W_k[\rho(t)] = -i\left(H\rho(t) - \rho(t)H\right) + e^{-k}L_1\rho(t)L_1^{\dagger} - \frac{1}{2}\{L_1^{\dagger}L_1, \rho(t)\}. \tag{17}$$

Here, $W_k$ is the so-called tilted Lindblad generator, which obeys $W_{k=0} = W$ [$W$ is defined in Eq. (1)]. Note, that unlike for the generator $W$, the propagation under (17), for $k \neq 0$, is not trace preserving and thus cannot represent a proper quantum dynamics. From Eq. (9) and

the definition of the density matrix $\rho(k, t)$ in Eq. (16), the following useful identity directly follows

$$\sum_{N=0}^{\infty} \mathcal{V}^{(0)}(N, t, t_0) e^{-kN} = e^{(t-t_0)W_k}. \tag{18}$$

In the case $k = 0$, the previous equation reduces to Eq. (10) as it must be. Combining Eq. (17) with the last equality in Eq. (15) and the definition of the SCGF in Eq. (13), one realizes that $\theta(k)$ coincides with the largest real eigenvalue of $W_k$, see Refs. [9, 12, 17, 19, 23, 24, 27] for a detailed discussion. In the case of Markovian-dissipative quantum systems, the calculation of the large deviation statistics of quantum-jump trajectories can be therefore accomplished by studying the spectrum of the tilted generator $W_k$. It is worth to mention here, that from the tilted generator $W_k$ in Eq. (17) one can construct a $k$-dependent physical, i.e., trace preserving, Lindblad generator which encondes as typical quantum-jump trajectories the rare trajectories of the original Lindblad generator $W$. This is accomplished via the quantum Doob transform introduced in Ref. [9, 17] and it is a crucial result as it allows for the numerical observation of trajectories at the tails of the activity distribution $P_t(N)$, which would otherwise be extremely hard to observe due to the exponential suppression in time in Eq. (12).

## 2.2 Stochastic resetting in open quantum systems: the non-Markovian generalized Lindblad equation

Stochastic resetting corresponds to the halting of the time evolution of the system at random times at which the system state is reset to a *reset state* $|\psi_r\rangle$. We emphasize that the probability of observing a reset does not depend on the state of the system. The system then evolves again in time according to its reset-free dynamics, which we denote as $\rho^{(0)}(t)$, until the next reset is performed. In this manuscript we consider the case where the reset-free time evolution $\rho^{(0)}(t)$ is Markovian and dissipative and it is thereby governed by a Lindblad generator $W^{(0)}$ as in Eq. (1):

$$\rho^{(0)}(t) = e^{W^{(0)}t}[\rho(0)], \quad \text{with} \quad \rho(0) = |\psi_r\rangle \langle\psi_r|. \tag{19}$$

Here we assumed, without loss of generality, that the initial state $\rho(0)$ of the system coincides with the reset state. The resetting dynamics is eventually characterized by the *waiting-time distribution* $p(\tau)$, which is the probability density of the time $\tau$ elapsed between consecutive resets. A related quantity is the survival probability $q(t)$

$$q(t) = \int_t^{\infty} d\tau\, p(\tau) = 1 - \int_0^t d\tau\, p(\tau), \tag{20}$$

which is the probability that no reset happens before a time $t$ has elapsed since the last reset.

The resetting dynamics corresponds to a so-called *renewal process*, see, e.g., Ref. [79], in the sense that the dynamics after a certain reset does not depend on what happened before that reset. On the basis of this observation one can derive renewal equations relating the dynamics of the density matrix $\rho(t)$ in the presence of resets to the reset-free one $\rho^{(0)}(t)$, where resets are by definition absent. This approach has been pursued in Refs. [75, 76] for unitary reset-free time evolution, while we consider here the case of an underlying dissipative dynamics as in Eq. (19). The renewal equation for the density matrix $\rho(t)$ reads

$$\rho(t) = q(t)\rho^{(0)}(t) + \int_0^t d\tau\, \nu(\tau)q(t-\tau)\rho^{(0)}(t-\tau). \tag{21}$$

The first term on the right hand side describes a realization of the evolution where no reset event happens before time $t$ and it is therefore given by the reset-free evolution weighted by the survival probability $q(t)\rho^{(0)}(t)$. The second term, on the other hand, describes realizations where the last reset happened at time $\tau$ with probability density $\nu(\tau)$ and then the system evolves without reset during the remaining time $t - \tau$. Equation (21) is named for this reason "last renewal equation" as it involves the time $\tau$ of the last reset. The function $\nu(t)$ denotes the probability density of observing a reset exactly at time $t$ and it can be expressed in terms of the waiting time density $p(\tau)$ exploiting the renewal structure of the process. Namely, the probability density $\nu_R(t)$ that the $R_{th}$ reset occurs at time $t$ is given by

$$\nu_R(t) = \int_0^t d\tau \, p(\tau)\nu_{R-1}(t - \tau), \tag{22}$$

which expresses that the $(R-1)_{th}$ reset happened at time $t - \tau$ and then a time $\tau$ elapsed before the next event. By definition $\nu_1(t) = p(t)$ and

$$\nu(t) = \sum_{R=1}^{\infty} \nu_R(t). \tag{23}$$

The recurrence relation (22) can be solved in the Laplace-$s$ domain since it is a convolution. Upon taking the Laplace transform in Eq. (23) one indeed finds

$$\widehat{\nu}(s) = \int_0^{\infty} dt \, \nu(t)e^{-st} = \sum_{R=1}^{\infty} \widehat{p}(s)^R = \frac{\widehat{p}(s)}{1 - \widehat{p}(s)}, \quad \text{with} \quad \widehat{\nu}_R(s) = \widehat{p}(s)^R, \tag{24}$$

where in the last step we summed the geometric series. Equation (21) allows for the determination of the dynamics of the density matrix $\rho(t)$ once the reset-free time evolution $\rho^{(0)}(t)$ is known.

In a complementary way, one can characterize the dynamics by writing an evolution equation for $\rho(t)$, which allows to highlight the relation between the emergent dynamics in the presence of reset and the open-dissipative dynamics introduced in Subsec. 2.1. This connection has been first carried out in Ref. [55] for classical systems and in Ref. [76] for a reset-free unitary quantum dynamics. The generalization to the dissipative dynamics in Eq. (19) carries no additional conceptual difficulty and it is therefore briefly reported in the Appendix A. In particular, Eq. (21) is equivalent to

$$\frac{d}{dt}\rho(t) = W^{(0)}[\rho(t)] + \nu(t) |\psi_r\rangle \langle \psi_r| - \int_0^t dt' r(t - t')e^{(t-t')W^{(0)}}[\rho(t')], \tag{25}$$

with the function $r(t)$ satisfying the relation

$$\nu(t) = \int_0^t dt' r(t'), \tag{26}$$

which ensures that the dynamics in Eq. (25) is trace preserving. The expression of $r(t)$ in Laplace space readily follows from Eq. (24):

$$\widehat{r}(s) = \frac{s\widehat{p}(s)}{1 - \widehat{p}(s)} = \frac{\widehat{p}(s)}{\widehat{q}(s)}. \tag{27}$$

Here, we used the definition of the survival probability $q(t)$ in Eq. (20). The first term on the right hand side of Eq. (25) describes the reset-free dissipative dynamics dictated by the Lindbladian $W$. The second term is the gain term at the reset state $|\psi_r\rangle$ with probability density $\nu(t)$. The last term on the right hand side is, instead, the resetting escape rate associated with realizations where the system evolves under the dissipative dynamics in Eq. (1) between time $t'$ and $t$ and it is then subject to resetting. Equation (25) has therefore the structure of a master equation and it is therefore dubbed in the classical realm generalized master equation [55]. In the present manuscript, in order to highlight the connection with the Lindblad dissipative dynamics, we refer to (25) as *generalized Lindblad equation*, in the same way as in Ref. [76]. In particular, Eq. (25) shows that stochastic resetting can be considered as an engineered dissipative process additional to the ones contained in the Lindbladian $W$, due to the interaction between the system and the environment. The similarities between Eq. (25) and the Lindblad dynamics will be analyzed in detail in Sec. 3, particularly in discussing the formulation of Eq. (25) in terms of the quantum trajectories, introduced in Subsec. 2.1.

The fundamental difference between the generalized Lindblad equation and the Lindblad dynamics in Eq. (1) is that the latter is Markovian, i.e., time-independent, while the former is non-Markovian since the left hand side of Eq. (25) contains the time-dependent rate $\nu(t)$ and an integral over the complete history of the process before time $t$. The dynamics ensuing from the resetting procedure is thus non-Markovian despite the reset-free open quantum dynamics in Eq. (1) being Markovian. Consequently, the large deviation statistics of the quantum-jump trajectories associated with a dissipative process $L_1$ cannot be simply analytically computed from the spectrum of the tilted generator introduced in Eq. (17). Furthermore, the Doob transform method of Ref. [9, 17] applies only to Markovian-Lindblad open quantum systems. As such, the latter method cannot be utilised for treating the non-Markovian Eq. (25).

In spite of these difficulties, in Sec. 3, we, however, show that the large deviation statistics of the quantum-jump trajectories can be exactly computed exploiting the renewal structure of the dynamics. Before doing this, we briefly discuss the particular case of Poissonian resetting, which is instructive since in this regime Eq. (25) reduces to the Lindblad dynamics.

## 2.3 Poissonian resetting: Lindblad dynamics

For the case of Poissonian resetting the time between consecutive resets is distributed according to an exponential waiting time distribution $p_\gamma(\tau)$

$$p_\gamma(\tau) = \gamma e^{-\gamma\tau}, \tag{28}$$

while the survival probability in Eq. (20) takes the form

$$q_\gamma(t) = e^{-\gamma t}. \tag{29}$$

The occurrence of reset events is hence described by a Poisson process, see, e.g., Ref. [80], and therefore resets happen at constant (i.e. time-independent) rate $\gamma$. The probability density $\nu(t)$ for a reset to happen at time $t$ in Eqs. (23) and (24) simplifies, therefore, for Eqs. (28) and (29) to $\nu_\gamma(t) = \gamma$. The function $r(t)$ in Eqs. (26) and (27) similarly simplifies to

$$\widehat{r}_\gamma(s) = \gamma, \quad \text{and} \quad r_\gamma = \gamma\,\delta(t). \tag{30}$$

Inserting Eq. (30) into Eq. (25) one obtains the Lindblad equation

$$\frac{\mathrm{d}}{\mathrm{d}t}\rho(t) = W^{(\gamma)}[\rho(t)] = W^{(0)}[\rho(t)] + \gamma\,\mathrm{Tr}[\rho(t)]\,|\psi_r\rangle\,\langle\psi_r| - \gamma\rho(t). \tag{31}$$

In the previous equation we defined the Lindbladian $W^{(\gamma)}$ in the presence of reset, which is obtained from the reset-free Lindbladian $W^{(0)}$ in Eq. (1) by adding $\dim(\mathcal{H})$ jump operators, where $\dim(\mathcal{H})$ is the dimension of the Hilbert space $\mathcal{H}$. These jump operators read

$$L_i^{(\gamma)} = \sqrt{\gamma} \, |\psi_r\rangle \langle \phi_i| \,, \quad \text{with} \quad i = 1, 2 \dots \dim(\mathcal{H}), \tag{32}$$

with the $|\phi_i\rangle$ forming an orthonormal basis, i.e., $\langle \phi_i | \phi_j \rangle = \delta_{ij}$ (see also Ref. [72]). Note that the loss term in Eq. (31) is just

$$\gamma = \sum_i^{\dim(\mathcal{H})} (L_i^{(\gamma)})^\dagger L_i^{(\gamma)} = \gamma \mathbb{I}, \tag{33}$$

which expresses the fact that every state attained by the system during the reset-free evolution is reset at constant rate $\gamma$. We therefore see that, due to the fact that in the Poisson process resets occur with a constant rate $\gamma$, the underlying dynamics is Markovian and described by a Lindblad equation. This means furthermore, as discussed after Eq. (25), that stochastic resetting can be considered as a dissipative process additional to the one mediated by the jump operator $L_1$ contained in the reset-free Lindbladian $W^{(0)}$. The effective Hamiltonian $H_{\text{eff}}^{(\gamma)}$ in the presence of reset, from Eq. (33), is simply a shift of the effective Hamiltonian $H_{\text{eff}}^{(0)}$ of the reset-free dynamics

$$H_{\text{eff}}^{(\gamma)} = H_{\text{eff}}^{(0)} - i \frac{\gamma}{2} \mathbb{I}. \tag{34}$$

It is therefore useful to define the conditioned density matrix $\rho(N, R, t)$ conditioned on $N \geq 0$ jumps given by $L_1$ and $R \geq 0$ resets. The density matrix $\rho(t)$ is obtained, see Eq. (8), as

$$\rho(t) = \sum_{N=0}^{\infty} \sum_{R=0}^{\infty} \rho(N, R, t). \tag{35}$$

The decomposition of $\rho(N, R, t)$ into quantum trajectories with $R$ resets happening at times $t_1 \leq t_2 \dots t_R \leq t$ and $N$ jumps in $(0, t)$ can be directly written from Eqs. (5) and (9) with $H_{\text{eff}}^{(\gamma)}$ in Eq. (34)

$$\rho(N, R, t) = \sum_{l_1=0}^{N} \dots \sum_{l_R=0}^{N - \sum_{i=1}^{R-1} l_i} \int_0^t \mathrm{d}t_R \int_0^{t_R} \mathrm{d}t_{R-1} \dots \int_0^{t_2} \mathrm{d}t_1 q_\gamma(t - t_R) \mathcal{V}^{(0)} \left( N - \sum_{i=1}^{R} l_i, t - t_R, 0 \right)$$
$$p_\gamma(t_R - t_{R-1}) \mathcal{R} \mathcal{V}^{(0)}(l_R, t_R - t_{R-1}, 0) \dots p_\gamma(t_1) \mathcal{R} \mathcal{V}^{(0)}(l_1, t_1, 0)[\rho(0)], \tag{36}$$

where the superoperator $\mathcal{V}^{(0)}(N, t, t_0)$ for the reset-free dynamics has been defined in Eq. (9). In the previous equation we further defined, following the notation in Eq. (4), the superoperator $\mathcal{R}$ as

$$\mathcal{R}[\rho] = \sum_{i=1}^{\dim(\mathcal{H})} R_i \rho R_i^\dagger = \mathrm{Tr}[\rho] \, |\psi_r\rangle \langle \psi_r| \,, \quad \text{with} \quad R_i = L_i^{(\gamma)}/\sqrt{\gamma} = |\psi_r\rangle \langle \phi_i| \,. \tag{37}$$

When acting on unit-trace matrices, the operator $\mathcal{R}$ is a projector onto the reset state as $\mathcal{R}^2[\rho] = \mathcal{R}[\rho] = |\psi_r\rangle \langle \psi_r|$. The initial state $\rho(0)$ is chosen equal to the reset state as in Eq. (19).

Note that the appearance of the survival probability $q_\gamma$ and the waiting time distribution $p_\gamma$, in Eqs. (29) and (28), respectively, directly follows from the constant term in the effective Hamiltonian in Eq. (34) and the definition of the reset jump operators in Eq. (37).

The physical interpretation of Eq. (36) becomes clear within the renewal structure enforced by stochastic resetting: the quantum trajectories of the system are split into $R$ independent trajectories of the reset-free process all starting from the reset state $\rho(0)$. The superoperator $\mathcal{V}^{(0)}(l_i, t_i - t_{i-1}, 0)$, with $i = 1, 2, \ldots R$, then governs the reset-free Lindblad evolution with $l_i$ jumps between the $(i-1)_{th}$ and the $i_{th}$ reset. The factor inside the integrals in the first line of Eq. (36) accounts for the fact that after the time $t_R$ of the $R_{th}$ reset, the system does not experience any further reset. In the corresponding reset-free dynamics $N - \sum_{i=1}^R l_i$ jumps take place, such that the total number of jumps in $(0, t)$ is exactly $N$. The density matrix $\rho(N, R, t)$ is eventually attained upon integrating over all the possible allocations of the reset times $t_i$ and summing over all the corresponding number of jumps $l_i$ ($i = 1, 2 \ldots R$). Note, that the sums over the number of jumps are constrained, with the sum over $l_i$ running from 0 to $N - \sum_{j=1}^{i-1} l_j$. This ensures that the number of jumps happening before the $R_{th}$ reset time is at most equal to $N$.

The Lindblad dynamics in Eqs. (31)-(37) further allows for the calculation of the large deviation statistics of the quantum-jump trajectories associated with $L_1$ with the tilted generator approach, introduced in Subsec. 2.1, upon looking at the largest real eigenvalue of the tilted generator $W_k^{(\gamma)}(\rho)$

$$W_k^{(\gamma)}[\rho] = W_k^{(0)}[\rho] + \text{Tr}(\rho) - \gamma\rho. \tag{38}$$

In the following Section we keep the case of Poissonian resetting as a reference, while we focus on the more challenging case of arbitrary waiting time distributions, where the SCGF cannot be obtained by diagonalizing the tilted generator. We show that, nevertheless, the large deviation statistics of the quantum-jump trajectories can still be exactly computed exploiting the renewal structure of the trajectories as in Eq. (36).

## 3 Large Deviations of quantum-jump trajectories for the non-Markovian generalized Lindblad equation

In this Section we present the main result of this work, which is the exact calculation of the large deviation statistics of quantum-jump trajectories associated with the generalized Lindblad dynamics of Eq. (25). In Subsec. 3.1, we derive the tilted non-Markovian generator corresponding to Eq. (25) starting from the master equation formulation in terms of quantum trajectories, introduced in Subsec. 2.1 and 2.3. In particular, we show that, despite the dynamics being non-Markovian, it is possible to exactly calculate the moment generating function in the presence of resets. This is achieved by relating it to the reset-free moment generating function, exploiting the renewal structure of the dynamics. In Subsec. 3.2, we show that the tilted generator can be equivalently derived, without making reference to the quantum-trajectories formulation of the master equation, directly starting from the last renewal equation for the tilted density matrix $\rho(k, t)$. This is achieved by a procedure analogous to the one introduced in Subsec. 2.2 for the density matrix $\rho(t)$. The main result for the calculation of the moment generating function and the associated SCGF is given in Eqs. (50)-(55). In this case, indeed, the tilted generator is time dependent and the SCGF cannot be computed from its spectrum.

Equations (50) and (55) therefore represent the only analytical method, to our knowledge, to compute the SCGF in the present context of the generalized Lindblad equation (25).

## 3.1 The non-Markovian tilted generator from quantum trajectories

The moment generating function $G^{(0)}(k, t)$ of the reset-free time evolution is defined as in Eq. (15)

$$G^{(0)}(k, t) = \text{Tr}[\rho^{(0)}(k, t)], \tag{39}$$

consistently with the notation introduced in Eq. (19). In the same way, $\rho^{(0)}(k, t)$ denotes the tilted density matrix for the reset-free tilted Lindblad dynamics in Eq. (17). The generating function $G(k, t)$ in the presence of resets is related to the tilted density matrix $\rho(k, t)$ by a relation identical to Eq. (39) for the reset-free process

$$G(k, t) = \sum_{N=0}^{\infty} P_t(N) e^{-kN} = \text{Tr}[\rho(k, t)]. \tag{40}$$

Henceforth in this manuscript, we shall always use the superscript (0) to distinguish quantities of the reset-free process from the corresponding ones in the reset dynamics (in the latter case no superscript is used). We want to determine here the dynamical equation ruling the time evolution of $\rho(k, t)$ with resets. To do this, we observe that, despite the dynamics in Eq. (25) being non-Markovian, the renewal structure of the resetting protocol still holds. Consequently, the conditional density matrix $\rho(N, R, t)$ can be written in terms of the quantum trajectories as in Eq. (36). One merely needs to replace the survival probability and the waiting time distribution of the Poissonian resetting with an arbitrary $q(t)$ and $p(t)$, respectively:

$$\rho(N, R, t) = \sum_{l_1=0}^{N} \cdots \sum_{l_R=0}^{N-\sum_{i=1}^{R-1} l_i} \int_0^t dt_R \int_0^{t_R} dt_{R-1} \cdots \int_0^{t_2} dt_1 q(t - t_R) \mathcal{V}^{(0)} \left( N - \sum_{i=1}^{R} l_i, t - t_R, 0 \right)$$
$$p(t_R - t_{R-1}) \mathcal{R} \mathcal{V}^{(0)}(l_R, t_R - t_{R-1}, 0) \dots p(t_1) \mathcal{R} \mathcal{V}^{(0)}(l_1, t_1, 0)[\rho(0)]. \tag{41}$$

The evolution equation for $\rho(N, R, t)$ is attained upon differentiating this equation with respect to $t$. The result is (the intermediate technical steps are reported in the Appendix B)

$$\frac{d}{dt} \rho(N, R, t) = \mathcal{L}_0[\rho(N, R, t)] + \mathcal{L}_1[\rho(N - 1, R, t)](1 - \delta_{N,0}) + \nu(N, R, t) |\psi_r\rangle \langle\psi_r| (1 - \delta_{R,0})$$
$$- \sum_{M=0}^{N} \int_0^t dt' r(t - t') \mathcal{V}^{(0)}(M, t, t')[\rho(N - M, R, t')]. \tag{42}$$

The first two terms on the right hand side of the first line account for the reset-free dynamics [see Eq. (11)]. The term $\nu(N, R, t)$ is the probability density that the $R_{th}$ reset happens exactly at time $t$ when exactly $N$ jumps associated with $L_1$ occurred in the interval $(0, t)$. This term is therefore the "probability gain" for the reset state $|\psi_r\rangle$ due to quantum trajectories with $N$ jumps and $R$ resets. Its expression in terms of the quantum trajectories is given by

$$\nu(N, R, t) = \sum_{l_1=0}^{N} \cdots \sum_{l_{R-1}=0}^{N-\sum_{i=1}^{R-2} l_i} \int_0^t dt_{R-1} \int_0^{t_{R-1}} dt_{R-2} \cdots \int_0^{t_2} dt_1 \, p(t - t_{R-1}) P_{t-t_{R-1}}^{(0)} \left( N - \sum_{i=1}^{R-1} l_i \right)$$
$$p(t_{R-1} - t_{R-2}) P_{t_{R-1} - t_{R-2}}^{(0)}(l_{R-1}) \dots p(t_1) P_{t_1}^{(0)}(l_1). \tag{43}$$

In the previous equation $P_t^{(0)}(N)$ denotes the probability of observing $N$ quantum jumps in the interval $(0, t)$ for the reset-free dynamics, according to Eqs. (7) and (9). Equation (43) therefore counts all the quantum trajectories ending with the $R_{th}$ reset at time $t$; hence, the factor $p(t - t_{R-1})$ on the first line of the equation, with exactly $N$ jumps before time $t$. The term in the second line of Eq. (42), instead, is the resetting escape rate associated with quantum trajectories where the system experiences $M$ quantum jumps in the time interval $t - t'$ and is then subject to resetting [$r(t)$ is defined in Eqs. (26) and (27)]. The function $\nu(N, R, t)$ can also be written in terms of the density matrix (see again the Appendix B for the details of the calculation) as

$$\nu(N, R, t) = \text{Tr}\left[\sum_{M=0}^{N} \int_0^t dt' r(t - t') \mathcal{V}^{(0)}(M, t, t')[\rho(N - M, R - 1, t')]\right]. \tag{44}$$

In order to derive the non-Markovian tilted generator corresponding to Eq. (42), we proceed by introducing the tilted density matrix $\rho(k, t)$ in the presence of resets with a discrete Laplace transform in the variable $N$

$$\rho(k, t) = \sum_{N=0}^{\infty} \sum_{R=0}^{\infty} e^{-kN} \rho(N, R, t) = \sum_{N=0}^{\infty} e^{-kN} \rho(N, t). \tag{45}$$

The series in Eq. (43) decouple upon summing over the number $N$ of jumps, due to the discrete convolution structure, and one obtains

$$\nu(k, t) = \sum_{R=1}^{\infty} \int_0^t dt_{R-1} \int_0^{t_{R-1}} dt_{R-2} \cdots \int_0^{t_2} dt_1 \, p(t - t_{R-1}) G^{(0)}(k, t - t_{R-1})$$
$$p(t_{R-1} - t_{R-2}) G^{(0)}(k, t_{R-1} - t_{R-2}) \dots p(t_1) G^{(0)}(k, t_1), \tag{46}$$

where we used Eq. (18) and then the definition in Eq. (39) for the reset-free moment generating function. Note that for $k = 0$ Eq. (46) reduces to Eqs. (22) and (23). The expression of $\nu(k, t)$ in terms of the density matrix $\rho(k, t)$ can be obtained from Eq. (44) as

$$\nu(k, t) = \text{Tr}\left[\int_0^t dt' r(t - t') e^{(t - t') W_k^{(0)}}[\rho(k, t')]\right], \tag{47}$$

where Eq. (18) has been used. Equation (47) unveils the generalized-Lindblad structure of the ensuing dynamics. Inserting Eq. (47) into Eq. (42), with the jump operators $R_i$ in Eq. (37), one obtains

$$\frac{\partial \rho(k, t)}{\partial t} = W_k^{(0)}[\rho(k, t)] + \sum_{i=1}^{\dim(\mathcal{H})} \int_0^t dt' r(t - t') R_i \left(e^{(t - t') W_k^{(0)}}[\rho(k, t')]\right) R_i^\dagger$$
$$-\frac{1}{2} \sum_{i=1}^{\dim(\mathcal{H})} \int_0^t dt' r(t - t') \left\{R_i^\dagger R_i, e^{(t - t') W_k^{(0)}}[\rho(k, t')]\right\}. \tag{48}$$

The previous equation generalizes the Lindblad equation (31) by integrating the density matrix over the entire dynamics before time $t$ with a memory kernel provided by the function $r(t - t')$. This is a consequence of the fact that when resetting does not happen at constant rate $\gamma$, i.e., for Poissonian resetting, one must always keep track of the time elapsed since the

last reset, which results into a non-Markovian dynamics. In the Poissonian case, the memory kernel $r_\gamma(t - t')$ in Eq. (30) gives back the Lindblad dynamics in Eq. (31). Furthermore, for $k = 0$, Eq. (48) reduces to the generalized Lindblad equation (25). Our derivation of Eq. (48) thereby provides the non-Markovian tilted generator for the dynamics of an open quantum system subject to stochastic resetting with an arbitrary waiting time distribution. We mention that for any $k \neq 0$, the evolution under the generator $W_k^{(0)}$ is not trace-preserving. However, using the quantum Doob transformation of Ref. [9, 17] one can derive an associated Doob-transformed Lindbladian whose typical trajectories correspond to rare trajectories of $W^{(0)}$. For the non-Markovian tilted generator in Eq. (48), however, the procedure of Ref. [9, 17] to derive the Doob transformed generator does not apply. The definition of the Doob transform for the non-Markovian tilted dynamics in Eq. (48) is an open and challenging question which, however, goes beyond the scope of the present manuscript. Equations non-local in time similar to Eq. (48) have been derived in Refs. [81–84] for classical semi-Markov systems, where transitions between different configurations depend on the time elapsed since the last jump and on the outcome of the latter but not on the previous history of the process.

The SCGF $\theta(k)$ in the presence of resets cannot be directly computed from Eq. (48), since the tilted generator is not time-independent. However we can compute $\theta(k)$ exploiting the renewal structure of the quantum trajectories in Eq. (41). Upon inserting the expression for $\rho(N, R, t)$ into the discrete Laplace transform (45) and taking the trace of both sides of the equality, one eventually gets

$$G(k, t) = \sum_{R=0}^{\infty} \int_0^t \mathrm{d}t_R \int_0^{t_R} \mathrm{d}t_{R-1} \cdots \int_0^{t_2} \mathrm{d}t_1 q(t - t_R) G^{(0)}(k, t - t_R)$$
$$p(t_R - t_{R-1}) G^{(0)}(k, t_R - t_{R-1}) \ldots p(t_1) G^{(0)}(k, t_1), \qquad (49)$$

with the definition of the moment generating function $G(k, t)$ in Eq. (40). Equation (49) allows for the calculation of the generating function $G(k, t)$ in the presence of resets solely on the basis of the knowledge of the generating function $G^{(0)}(k, t)$ of the reset-free process. An equation analogous to Eq. (49) has been derived using a renewal argument for time-integrated observables in classical Markov processes subject to Poissonian resetting in Refs. [66, 67]. In the present case, Eq. (49) is derived directly from the renewal structure of the quantum trajectories (41) of Markovian open quantum systems subject to stochastic resetting with an arbitrary waiting time distribution. To compute the SCGF $\theta(k)$ we then proceed as in Refs. [66, 67] by introducing the Laplace transform $\widehat{G}(k, s)$ of $G(k, t)$ with respect to time [2]

$$\widehat{G}(k, s) = \int_0^{\infty} \mathrm{d}t\, G(k, t) e^{-st}, \qquad (50)$$

where $s \in \mathbb{C}$ is a complex variable. The Laplace transform of Eq. (49), upon recognizing the convolution structure of the $R$ integrals involved, gives

$$\widehat{G}(k, s) = \sum_{R=0}^{\infty} \widehat{Q}(k, s) \widehat{P}(k, s)^R = \frac{\widehat{Q}(k, s)}{1 - \widehat{P}(k, s)}, \qquad (51)$$

provided $|\widehat{P}(k, s)| < 1$ and the geometric series therefore converges. The functions $\widehat{P}(k, s)$

---

[2]The convention used in this manuscript is different from the one mostly used in the literature, where $s$ is the variable conjugated to $N$ in the discrete Laplace transform in Eq. (16).

and $\widehat{Q}(k,s)$ are given by

$$\widehat{P}(k,s) = \int_0^\infty \mathrm{d}t\, p(t)\, G^{(0)}(k,t)e^{-st}, \tag{52}$$

and

$$\widehat{Q}(k,s) = \int_0^\infty \mathrm{d}t\, q(t)\, G^{(0)}(k,t)e^{-st}, \tag{53}$$

respectively. The SCGF $\theta(k)$ is a real function since $G(k,t)$ is real and non-negative because of its very definition in Eq. (15). In particular, $\theta(k)$ can be determined from Eq. (51) by observing that the large deviation principle in Eq. (13) (provided it applies as well in the presence of resets) in Laplace space reads

$$\widehat{G}(k,s) \sim \frac{1}{s - \theta(k)}. \tag{54}$$

The SCGF $\theta(k)$ in the presence of reset can be therefore obtained by locating the largest simple and real pole $s^*(k)$ of $\widehat{G}(k,s)$ in Eq. (51) (smaller poles give a sub-leading contribution to the long-time asymptotics of $G(k,t)$). Note that this can be accomplished by looking for the largest simple and real zero $s^*(k)$ of the denominator of Eq. (51)

$$\theta(k) = s^*(k) : 1 - \widehat{P}(k, s^*(k)) = 0, \tag{55}$$

provided the numerator $\widehat{Q}(k,s)$ is correspondingly finite. In Sec. 4, we compute the SCGF $\theta(k)$ numerically according to the method of Eqs. (51)-(55). Before presenting this, we show in the next Subsection how to obtain the tilted non-Markovian generator in Eq. (48) and the renewal equation for $G(k,t)$ in a different way, by directly writing the last renewal equation for $\rho(k,t)$, without reference to the quantum-trajectories interpretation in Eq. (41).

## 3.2 Renewal equation approach for the moment generating function

Renewal equations, see, e.g., Ref. [46, 57–63, 85], allow to immediately relate the probability distribution of the reset process to the reset-free one. We apply this approach here to the case of open quantum systems by relating the probability $P_t(N)$ of having $N$ jumps in the time interval $(0,t)$ in the presence of resets to the reset-free jump distribution $P_t^{(0)}(N)$ as

$$P_t(N) = q(t)P_t^{(0)}(N) + \sum_{M=0}^\infty \int_0^t \mathrm{d}t_1 p(t_1) P_{t_1}^{(0)}(M) P_{t-t_1}(N-M). \tag{56}$$

The factor $P_{t_1}^{(0)}(M)$ thereby accounts for the number $M$ of jumps occurring under the reset-free dynamics before the first reset at time $t_1$. The factor $P_{t-t_1}(N-M)$ accounts for the number $N-M$ of jumps taking place in the remaining time interval $t-t_1$ subject to the resetting dynamics. The sum over $M$ eventually counts all the possible way of distributing the quantum jumps in the time intervals $(0,t_1)$ and $(t_1,t)$ such that the total number of jumps in $(0,t)$ is equal to $N$. The structure of the previous equation is similar to the one of Eq. (21), except that it is written in terms of the time $t_1$ of the first reset. For this reason, we shall refer to Eq. (56) as "first renewal equation". Taking the discrete Laplace transform (40) of

Eq. (56), one directly obtains the first renewal equation for the generating function $G(k, t)$ of the reset process

$$G(k, t) = q(t)G^{(0)}(k, t) + \int_0^t \mathrm{d}t_1 G^{(0)}(k, t_1)G(k, t - t_1). \tag{57}$$

The Laplace transform in time $\widehat{G}(k, s)$ of the previous equation yields Eq. (51), as also noted in Ref. [67] in the context of Poissonian resetting in classical Markov systems.

One can, equivalently, write the last renewal equation for $G(k, t)$ in terms of the time $\tau$ of the last reset, as done in Eq. (21) for the density matrix $\rho(t)$. This yields

$$G(k, t) = q(t)G^{(0)}(k, t) + \int_0^t \mathrm{d}\tau \nu(k, \tau)q(t - \tau)G^{(0)}(k, t - \tau). \tag{58}$$

The function $\nu(k, \tau)$ must be defined such that the last (58) and the first (57) renewal equation render the same solution for $G(k, t)$ and they are therefore equivalent. This equivalence can be established by taking the Laplace transform of the last renewal equation (58). In order for the last and first renewal equations for $G(k, t)$ to be equivalent, one finds that the Laplace transform of $\nu(k, t)$ must be given by

$$\widehat{\nu}(k, s) = \frac{\widehat{P}(k, s)}{1 - \widehat{P}(k, s)}, \tag{59}$$

with $\widehat{P}(k, s)$ defined in Eq. (52). It is also immediate to see that the Laplace transform of the convolution integrals in Eq. (46) for $\nu(k, t)$ is equal to Eq. (59) thereby showing the equivalence between the renewal approach of this Subsection with the quantum-trajectories formulation of Subsec. 3.1.

From the last renewal equation (58) one can write, following the definition in Eq. (40), the corresponding equation for the tilted density matrix:

$$\rho(k, t) = q(t)\rho^{(0)}(k, t) + \int_0^t \mathrm{d}\tau \nu(k, \tau)q(t - \tau)\rho^{(0)}(k, t - \tau). \tag{60}$$

We emphasize that for $k = 0$ the previous equation coincides with Eq. (21) for $\rho(t)$. Equation (60) can be therefore considered as the last renewal equation for the tilted density matrix $\rho(k, t)$. Note that in passing from Eq. (21) to Eq. (60) not only the density matrix $\rho(k, t)$ (and $\rho^{(0)}(k, t)$) is tilted, but also the reset probability density $\nu(k, t)$, according to Eq. (59). Equation (60) can be rewritten in a differential form ( the derivation is very similar to the one leading from Eq. (21) to Eq. (25), cf. the Appendix A) leading to

$$\frac{\partial \rho(k, t)}{\partial t} = W_k^{(0)}[\rho(k, t)] + \nu(k, t)|\psi_r\rangle\langle\psi_r| - \int_0^t \mathrm{d}t' r(t - t')e^{(t - t')W_k^{(0)}}[\rho(k, t')], \tag{61}$$

which is equivalent to Eq. (48) once $\nu(k, t)$ is rewritten as in Eq. (47) and the reset jump operators $R_i$ in Eq. (37) are introduced. We have therefore proved that the moment generating function and the tilted non-Markovian generator of the reset dynamics can be directly derived from the renewal equation of $\rho(k, t)$. The thermodynamics of quantum-jump trajectories formulation of Subsec. 3.1 provides, however, a physically deeper approach than the one followed in this Subsection, since it explains the actual physical description of the dynamics highlighting the connection with the physics of dissipative systems.

# 4    Applications to quantum optics systems

In this Section we apply the theory developed in Sec. 3 to simple quantum optical systems. We consider, in particular, in Subsec. 4.1 the statistics of the number of emitted photons emitted from a driven two-level atomic system with incoherent decay. We show that the activity (photon emission rate) of the system can be either increased or reduced with respect to the absence of resetting, depending on the choice of the reset state $|\psi_r\rangle$. In Subsec. 4.2, we consider a three-level atomic system in the electron shelving configuration [7, 8], where the quantum-jump trajectories display "blinking" or intermittent behavior. The competition between the resetting and the electron shelving mechanism gives, in this case, rise to a dynamical behavior that is much richer than the one of the two-level system.

We mention that the characterization of the photon counting statistics, that we here perform exactly, is challenging from the numerical perspective as rare tails of the activity distribution are exponentially suppressed as a function of time (12). Moreover, the application of our results, given by Eqs. (50)-(55), to two and three-level atomic systems represents the first analytical prediction for the photon counting statistics of such systems in the presence of non-Poissonian resetting. In the latter case, as explained thoroughly in the Introduction 1 and in Sec. 3, it is not possible to obtain the SCGF from the spectrum of tilted generator in Eq. (48), since the latter is time-dependent.

## 4.1    Two-level atomic system

We consider a driven two-level system with Hamiltonian

$$H = \omega(\sigma^+ + \sigma^-), \tag{62}$$

with $\sigma^+ = |1\rangle\langle0|$ and $\sigma^- = |0\rangle\langle1|$ being the raising and lowering operators, respectively. The states $|0\rangle$ and $|1\rangle$ denote the two atomic levels. In the previous equation $\omega$ denotes the Rabi frequency. The dissipative part of the reset-free Lindblad dynamics in Eq. (1) is given by one jump operator, $N_L = 1$, which expresses the incoherent decay from the state $|1\rangle$ to $|0\rangle$

$$L_1 = \sqrt{\mathcal{K}}\sigma^-, \tag{63}$$

with decay rate $\mathcal{K}$. Here we consider the case $\mathcal{K} = 4\omega$ to simplify the calculations and since this choice has been shown in Refs. [9, 11] to induce an interesting self-similarity in the quantum-trajectory statistics. The number of detected jumps associated with $L_1$ corresponds to the number of emitted (and detected) photons into the environment. We henceforth, accordingly, refer to $|1\rangle$ as emitting or active state and to $|0\rangle$ as non-emitting or inactive state. One notices that the specific definition of $L_1$ in Eq. (63) defines a renewal process already at the level of the Lindblad equation, since the non-Hermitian evolution given by $H_{\text{eff}}$ in Eq. (3) is projected to the state $|0\rangle$ every time a jump given by $L_1$ takes place. This renewal process is, however, non-Poissonian as noted in Ref. [9].

We consider here the case where stochastic resetting is superimposed to the two-level dynamics. In particular, in order to explore the ensuing non-Markovian generalized Lindblad dynamics detailed in Secs. 3, we consider the case of a non-Poissonian waiting time distribution $p_{\gamma,t_{\max}}(\tau)$ defined as

$$p_{\gamma,t_{\max}}(\tau) = \frac{\gamma}{1 - \exp(-\gamma t_{\max})}e^{-\gamma\tau}\Theta(t_{\max} - \tau), \quad \text{with} \quad 0 < \tau < \infty, \tag{64}$$

with the associated survival probability

$$q_{\gamma,t_{\max}} = \left( \frac{e^{-\gamma\tau} - e^{-\gamma t_{\max}}}{1 - e^{-\gamma t_{\max}}} \right) \Theta(t_{\max} - \tau), \quad \text{with} \quad 0 < \tau < \infty. \tag{65}$$

The waiting time distribution $p_{\gamma,t_{\max}}(\tau)$ represents an exponential distribution, with rate $\gamma$, "chopped" at a maximum reset time $t_{\max}$. The choice of such waiting time distribution is motivated by the fact that it is experimentally easier to implement distributions that feature a finite maximum time $t_{\max}$ between consecutive resets. This is in contrast to the case of the purely exponentially decaying $p_\gamma(\tau)$ in Eq. (28) which allows for arbitrary long times between consecutive resets. Furthermore, from the theoretical perspective, $p_{\gamma,t_{\max}}(\tau)$ allows to recover the exponential distribution $p_\gamma(\tau)$ in the limit $t_{\max} \to \infty$. This allows to benchmark our results for the non-Markovian generalized Lindblad dynamics, in the limit $t_{\max} \to \infty$, with the Poissonian Lindblad dynamics of Subsec. 2.3.

In Fig. 2 we report the result for $\theta(k)$ (computed according to Eqs. (51)-(55)) and the associated rate function $I(n)$ (as in Eq. (14)) for the case the reset state is chosen as the non-emitting state: $|\psi_r\rangle = |0\rangle$. In panel (a), we see that the SCGF approaches the limiting shape given by the Poissonian resetting as $t_{\max}$ is increased. In the same limit one recovers the Lindblad equation and therefore $\theta(k)$ can be computed as the largest real eigenvalue of the tilted generator $W_k^{(\gamma)}$ in Eq. (38). Our analysis thereby allows for the calculation of $\theta(k)$ in the far more complicated case of arbitrary waiting time distributions, where the ensuing dynamics in Eqs. (48) is non-Markovian and the tilted generator cannot be diagonalized.

In panel (b) of Fig. 2 we show the large deviation function $I(n)$. From the plot it is clear that stochastic resetting, in this case, fosters the population of the non-emitting state thereby leading to a reduction of the activity with respect to the reset-free dynamics. This is witnessed both in typical trajectories and in rare-atypical fluctuations. The former are described by the average activity $\langle n \rangle_0 = -\theta'(k = 0)$, where the subscript in $\langle n \rangle_0$ refers to the choice of the reset state. The latter are, instead, captured by the tails of the large deviation function $I(n)$, which shows, indeed, a stronger and stronger suppression of strongly active trajectories, i.e., such that $n \gg \langle n \rangle_0$, as $t_{\max}$ is lowered and reset events become more frequent. Inactive trajectories, i.e., such that $n \ll \langle n \rangle_0$, become, in a complementary way, more likely to happen as $t_{\max}$ decreases.

In Fig. 3, one, indeed, immediately realizes that the average activity $\langle n \rangle_0$ is a monotonically increasing function of $t_{\max}$. In the limit $t_{\max} \to 0$, one encounters the quantum Zeno effect [86, 87], where the state of the system coincides with the reset state due to extremely frequent reset projections. Emissions are therefore absent since resetting here projects to the inactive state. In the Markovian limit, $t_{\max} \to \infty$, of Poissonian resetting at rate $\gamma$, the average activity approaches a limiting value, which is smaller than the average activity of the corresponding reset-free dynamics.

In Fig. 4, we report the results for $\theta(k)$ and the corresponding rate function $I(n)$ for the case the reset state is chosen as the emitting state: $|\psi_r\rangle = |1\rangle$. Stochastic resetting in this case fosters the population of the emitting state and therefore makes emissions plentiful. This aspect is reflected both in the atypical and the typical trajectories in the opposite way with respect to Fig. 2. In particular, strongly active trajectories become now more probable as $t_{\max}$ is lowered.

Typical trajectories are described by the average activity $\langle n \rangle_1$, which is plotted in Fig. 3. The mean activity is now larger than the corresponding one in the reset-free dynamics for

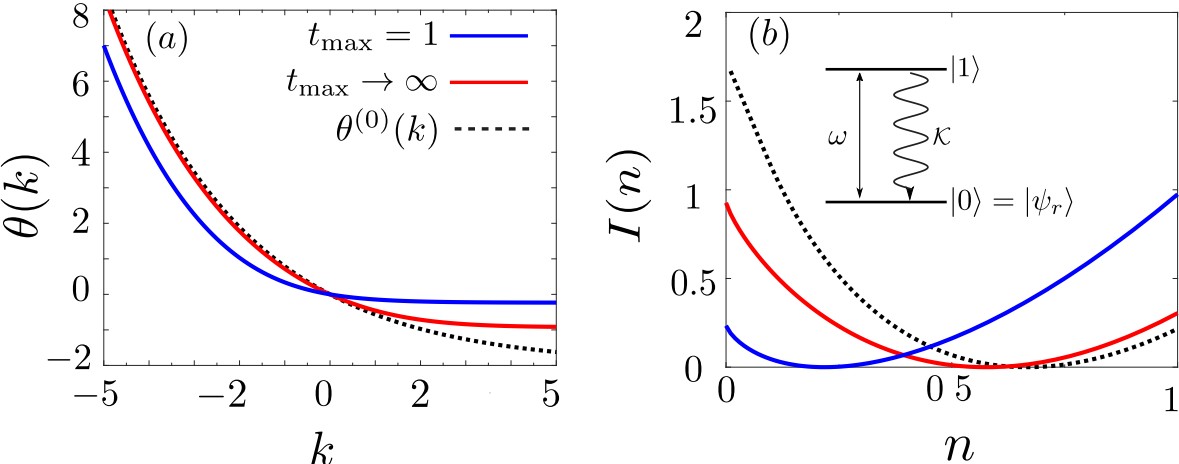

Figure 2: **Two-level system reset to the inactive state** $|\psi_r\rangle = |0\rangle$. (a) Scaled cumulant generating function $\theta(k)$ as a function of $k$ for $t_{\max} = 1$ (in units of $\omega^{-1}$, blue solid line) and $t_{\max} \to \infty$ (red solid line). The SCGF of the reset-free dissipative process $\theta^{(0)}(k)$ is shown as a black dashed curve for comparison. (b) Rate function $I(n)$ as a function of the activity $n = N/t$ for the same values of $t_{\max}$ as in panel (a). The two-level system is also sketched. Resetting favours the population of the inactive state $|0\rangle$ and therefore lowers the activity of the system. In the Figure we set $\gamma = 0.3$ and $\mathcal{K} = 4$ in units of $\omega$.

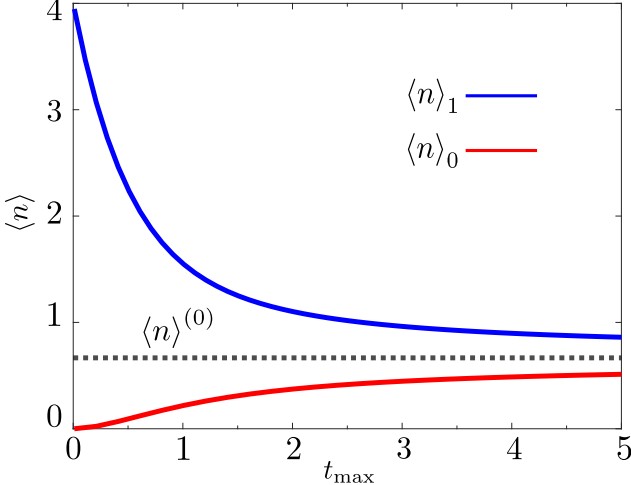

Figure 3: **Mean activity as a function of $t_{\max}$.** The latter is measured in units of $\omega^{-1}$. The top (blue) curve corresponds to the mean activity $\langle n \rangle_1$ in the case $|\psi_r\rangle = |1\rangle$, while the bottom (red) one corresponds to the mean activity $\langle n \rangle_0$ for the choice $|\psi_r\rangle = |0\rangle$. The dashed horizontal line $\langle n \rangle^{(0)}$ is the mean activity associated with the reset-free SCGF $\theta^{(0)}(k)$. In the Figure we set $\gamma = 0.3$ and $\mathcal{K} = 4$ in units of $\omega$.

every value of $t_{\max}$. We stress that as $t_{\max} \to 0$, in the quantum Zeno limit, $\langle n \rangle_1$ is, in this

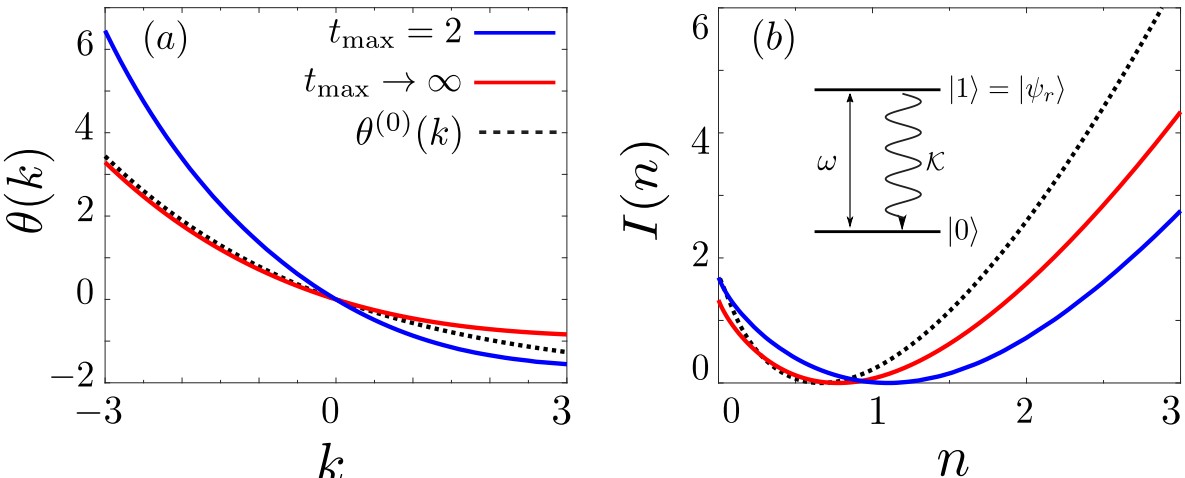

Figure 4: **Two-level system reset to the active state $|\psi_r\rangle = |1\rangle$.** (a) Scaled cumulant generating function $\theta(k)$ as a function of $k$ for $t_{\max} = 2$ (in units of $\omega^{-1}$, blue solid line) and $t_{\max} \to \infty$ (red solid line). The SCGF of the reset-free dissipative process $\theta^{(0)}(k)$ is shown as a black dashed curve for comparison. (b) Rate function $I(n)$ as a function of the activity $n = N/t$ for the same values of $t_{\max}$ as in panel (a). The two-level system is also sketched. Resetting favours the population of the active state $|1\rangle$ and therefore, contrarily to Fig. 2, enhances the activity of the system. In the Figure we set $\gamma = 0.3$ and $\mathcal{K} = 4$ in units of $\omega$.

case, non-zero since resetting projects to the emitting state. In particular we have

$$\langle n \rangle_1 = \text{Tr}[L_1^\dagger L_1 \rho] \to \left\langle \psi_r \left| L_1^\dagger L_1 \right| \psi_r \right\rangle = \left\langle 1 \left| L_1^\dagger L_1 \right| 1 \right\rangle = \mathcal{K}, \quad \text{as} \quad t_{\max} \to 0. \qquad (66)$$

In conclusion we can see that upon choosing the reset state $|\psi_r\rangle$ and the maximum reset time $t_{\max}$ (and the rate $\gamma$) one can tailor the activity of the system and its statistical fluctuations deep from the active to the inactive regime.

In the following Subsection we consider a three-level system, where a significantly richer behavior than the one described here can be observed upon introducing stochastic resetting.

## 4.2 Three-level atomic system

We now analyze a three-level system, whose Hamiltonian reads as

$$H = \sum_{j=1}^{2} \omega_j (\sigma_j + \sigma_j^\dagger) \qquad (67)$$

with $\sigma_j = |0\rangle \langle j|$. In the previous equation $\omega_1$ and $\omega_2$ denote the Rabi frequencies on the transition lines $|0\rangle - |1\rangle$ and $|0\rangle - |2\rangle$, respectively. As in the case of the two-level system of Subsec. 4.1, there is one jump operator only, $N_L = 1$ in the reset-free dynamics, which gives the incoherent decay from the state $|1\rangle$ to $|0\rangle$

$$L_1 = \sqrt{\mathcal{K}_1}\sigma_1 = |0\rangle \langle 1|, \qquad (68)$$

with rate $\mathcal{K}_1$. The detection of a quantum jump amounts therefore to the detection of a photon emitted into the environment on the transition line $|0\rangle - |1\rangle$. We consider the system in the "shelving" configuration, see, e.g., Refs. [7, 8], where $\omega_1 \gg \omega_2$. In this case the atom can be shelved for long times in the state $|2\rangle$ thereby strongly suppressing emissions on the $|1\rangle - |0\rangle$ line. The associated quantum-jump trajectories are intermittent or blinking as active periods, where emissions are plentiful, alternate with inactive periods, where emissions are scarce. This kind of dynamics has been interpreted in Ref. [9, 16] as a coexistence between an active and an inactive dynamical phase. Inactive periods can be, in particular, considered as "time bubbles" of the inactive phase into the active one, as shown in Ref. [16]. Intermittancy of the quantum-jump trajectories can be also observed in many-body open systems such as the dissipative transverse-field Ising chain analyzed in Ref. [12]. Therein, as a matter of fact, it has been shown that upon increasing the transverse field the quantum-jump statistics displays a dynamical transition from an inactive to an active dynamical phase. For intermediate values of the transverse field, instead, intermittency in the quantum-jump trajectories is observed. The transition between such dynamical phases is smooth as long as the number of spins in the chain is finite, since dynamical phases and transitions thereof exist in many-body systems only in the thermodynamic limit. The dynamics obtained for the three-level system therefore resembles the inactive-active dynamical transition of the dissipative transverse field Ising model as long as the number of spins in the chain is finite. It is also closely linked to dynamically metastable behvior as discussed in Refs. [88, 89].

In this Subsection we consequently explore the effect of the introduction of stochastic resetting on top of the aforementioned shelving dynamics. We take henceforth the waiting time distribution as the chopped exponential distribution $p_{\gamma,t_{\max}}(\tau)$ in Eq. (64). In Fig. 5 the results are shown for the SCGF $\theta(k)$ and the associated rate function $I(n)$ in the case the reset state $|\psi_r\rangle = |0\rangle$ is chosen as the inactive state of the $|1\rangle - |0\rangle$ line. In panel (a) we show that the SCGF $\theta(k)$ approaches the limiting shape given by Poissonian resetting, which can be computed as the largest real eigenvalue of the Markovian tilted generator $W_k^{(\gamma)}$ in Eq. (38), in the limit $t_{\max} \to \infty$. This further confirms the generality of our method to calculate the large deviation statistics, which applies to arbitrary waiting time distributions with an associated non-Markovian tilted generator (48). The slope of the reset-free SCGF (in black dashed in the Figure) $n^{(0)}(k) = -\mathrm{d}\,\theta^{(0)}(k)/\mathrm{d}k$ displays a sharp change around $k = 0$, which has been interpreted in Ref. [9] as a smoothened first-order crossover between the dynamical active and inactive phase. This crossover is reduced after the introduction of stochastic resetting. The physical interpretation is clear as resetting has the tendency of hindering the presence of long inactive periods, where the system is shelved into the state $|2\rangle$ and it is inactive, by re-initializing the system into the state $|0\rangle$.

In panel (b) of Fig. 5 we plot the associated rate function $I(n)$. The rate function of the corresponding reset-free dynamics given by Eqs. (67) and (68) is shown in black dashed. We can see that the qualitative behavior displayed by $I(n)$ as $t_{\max}$ changes is different than the corresponding one in Fig. 2 for the two-level system. This difference can be again realized both in atypical-rare trajectories and in typical ones. In the former case, indeed, the tails of the large deviation function $I(n)$ show that both strongly active trajectories, such that $n \gg \langle n \rangle_0$, and strongly inactive ones, such that $n \ll \langle n \rangle_0$, get suppressed with respect to the reset-free dynamics as $t_{\max}$ is reduced. The fat tail displayed for $n \ll \langle n \rangle_0$ by the reset-free large deviation function $I(n)$, which is a consequence of the sharp crossover of $\theta^{(0)}(k)$ around $k = 0$, is indeed lifted upon introducing the resetting protocol. The probability distribution

$P_t(n)$ of the activity therefore concentrates in a (approximately) symmetric way with respect to the average value $\langle n \rangle_0 = -\theta'(k=0)$.

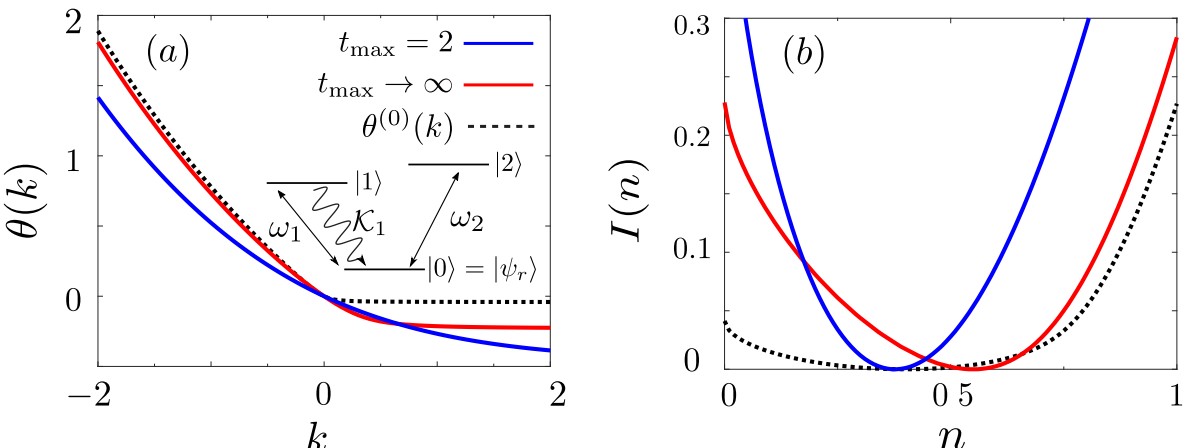

Figure 5: **Three-level system reset to the inactive state $|\psi_r\rangle = |0\rangle$.** (a) Scaled cumulant generating function $\theta(k)$ as a function of $k$ for $t_{\max} = 2$ (in units of $\omega_1^{-1}$, blue solid line) and $t_{\max} \to \infty$ (red solid line). The SCGF of the reset-free dissipative process $\theta^{(0)}(k)$ is shown as a black dashed curve for comparison. The three-level system is also sketched. (b) Rate function $I(n)$ as a function of the activity $n = N/t$ for the same values of $t_{\max}$ as in panel (a). Resetting hinders long inactive periods and therefore the intermittence of the quantum-jump trajectories. The crossover of $\theta(k)$ around $k = 0$ gets therefore smoother and smoother as reset events become more and more frequent (low values of $t_{\max}$). The large deviation function $I(n)$ accordingly becomes more and more symmetric with respect to the mean value $\langle n \rangle$ as $t_{\max}$ is lowered. In the Figure we set $\gamma = 0.2$, $\mathcal{K}_1 = 4$, $\omega_2 = 1/10$ in units of $\omega_1$.

The latter characterizes typical trajectories and it is plotted in Fig. 6 as a function of $t_{\max}$, again for the case $|\psi_r\rangle = |0\rangle$ of the reset state being equal to the non-emitting state. It is a monotonically increasing function, as in the case of the two-level system depicted in Fig. 3. In contrast to the latter case, however, $\langle n \rangle_0$ becomes larger than the reset-free activity $\langle n \rangle^{(0)}$ for sufficiently large values of $t_{\max}$. This behavior is quite counter intuitive as one would expect that resetting in this case hinders the activity of the system as it enhances the population of the non-emitting state $|0\rangle$. The solution of this conundrum lies in the fact that resetting allows the system to escape from the shelving state $|2\rangle$, where it can get trapped for long times, in favour of the state $|0\rangle$. From the latter state the system can then populate back the active state $|1\rangle$ thus enhancing the number of detected emissions. The competition between the maximum reset time $t_{\max}$ (or the resetting rate $\gamma$) and the shelving mechanism can therefore lead to an increase of the activity of the system with respect to the reset-free case, provided $t_{\max}$ is sufficiently large. In the opposite limit, as a matter of fact, of small values of $t_{\max}$, the activity is sharply reduced because of the quantum-Zeno effect, as in the case of Fig. 3.

In Fig. 7 we eventually consider the case where the reset state $|\psi_r\rangle = |2\rangle$ coincides with the state $|2\rangle$. In this case resetting fosters the population of the state $|2\rangle$ and therefore the intermittent or blinking nature of the quantum-jump trajectories. The crossover shown by the SCGF $\theta(k)$ around $k = 0$ accordingly becomes more pronounced than in the reset-free case, as shown in panel (a). The corresponding large deviation function $I(n)$, in panel (b),

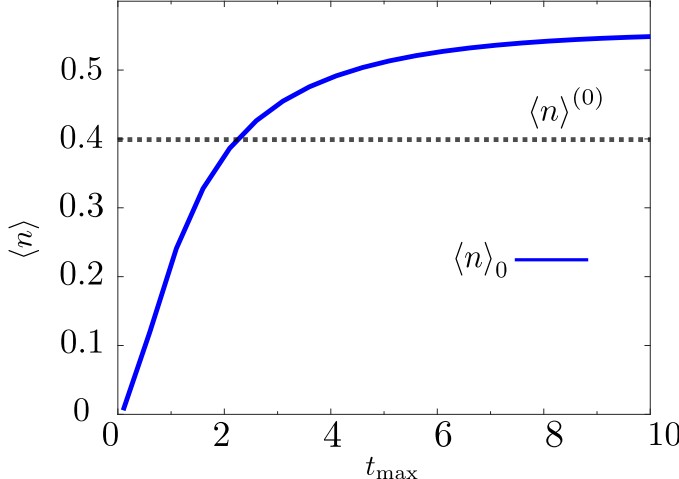

Figure 6: **Mean activity $\langle n \rangle_0$ as a function of $t_{\max}$ for the reset state $|\psi_r\rangle = |0\rangle$.** The maximum reset time $t_{\max}$ is measured in units of $\omega_1^{-1}$. The dashed horizontal line $\langle n \rangle^{(0)}$ is the mean activity associated with the reset-free SCGF $\theta^{(0)}(k)$. For sufficiently large values of $t_{\max}$ the system becomes more active than in the reset-free because resetting hinders the shelving into the state $|2\rangle$. In the limit of extremely small maximum reset time, instead, $t_{\max} \to 0$, the system is continuously projected to the inactive state $|0\rangle$ and the activity is zero. In the Figure we set $\gamma = 0.2$, $\mathcal{K}_1 = 4$, $\omega_2 = 1/10$ in units of $\omega_1$.

thereby peaks in a more and more pronounced asymmetric way with respect to the average value $\langle n \rangle_2$ as $t_{\max}$ is lowered.

In conclusion we can see that also in the case of the three-level system stochastic resetting can be exploited to engineer the activity of the system depending on the values of the maximum reset time $t_{\max}$ (and the rate $\gamma$). Furthermore the choice of the reset state $|\psi_r\rangle$ allows to tune the sharpness of the dynamical crossover between the dynamical phases thereby allowing for the control of the intermittence of the system's quantum-jump trajectories.

## 5 Summary and conclusion

In this manuscript we have developed a theory for the exact computation of the large deviation statistics of the quantum-jump trajectories in Markovian open quantum systems subject to stochastic resetting. We have first shown that the introduction of stochastic resetting on top of a Lindblad dynamics leads to an effective non-Markovian dissipative dynamics with the form of the generalized Lindblad equation in Eq. (25). Only in the case in which resetting takes place at a time-independent rate $\gamma$ — Poissonian resetting reviewed in Subsec. 2.3 — the generalized Lindblad equation reduces to the Lindblad equation (31). We show that despite the emerging dynamics in Eq. (25) being non-Markovian, the quantum-jump statistics can in fact be exactly computed. This is achieved by deriving, within the thermodynamics of quantum-jump trajectories formalism, a renewal equation, Eq. (49), which relates the generating function $G(k,t)$ of the activity in the presence of reset to the one $G^{(0)}(k,t)$ of the reset-free dynamics. The calculation of the SCGF $\theta(k)$ in the presence of resetting then

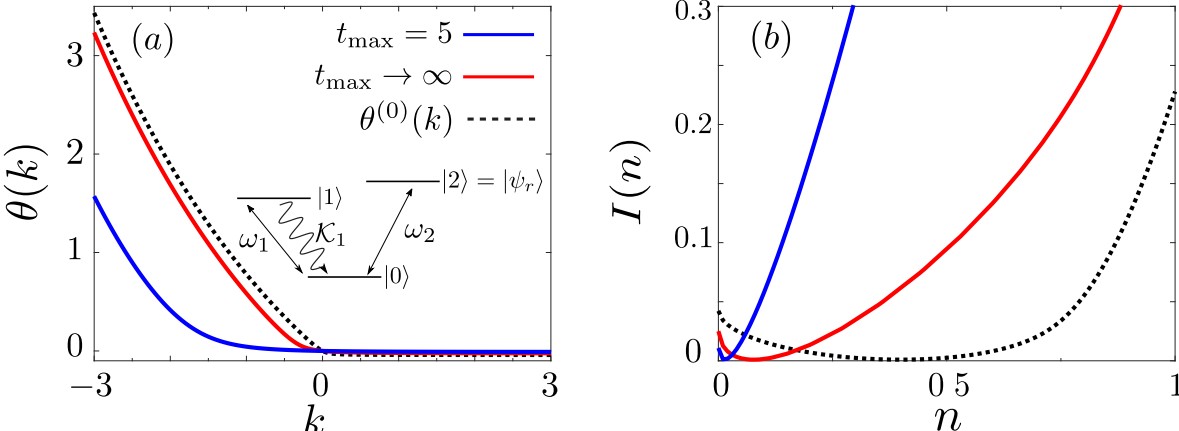

Figure 7: **Three-level system reset to the shelving state $|\psi_r\rangle = |2\rangle$.** (a) Scaled cumulant generating function $\theta(k)$ as a function of $k$ for $t_{\max} = 5$ (in units of $\omega_1^{-1}$, blue solid line) and $t_{\max} \to \infty$ (red solid line). The SCGF of the reset-free dissipative process $\theta^{(0)}(k)$ is shown as a black dashed curve for comparison. The three-level system is also sketched. (b) Rate function $I(n)$ as a function of the activity $n = N/t$ for the same values of $t_{\max}$ as in panel (a). Resetting favours long inactive periods and therefore the intermittence of the quantum-jump trajectories, in contrast to the case of Fig. 5. The crossover of $\theta(k)$ around $k = 0$ gets therefore sharper and sharper as reset events become more and more frequent (low values of $t_{\max}$). The large deviation function $I(n)$ accordingly becomes strongly asymmetric with respect to the mean value $\langle n \rangle$ as $t_{\max}$ is lowered. In the Figure we set $\gamma = 0.2$, $\mathcal{K}_1 = 4$, $\omega_2 = 1/10$ in units of $\omega_1$.

proceeds according to Eqs. (50)-(55), following the method developed in Refs. [66, 67] for classical Markovian systems subject to Poissonian resetting.

Our main result in Eqs. (50)-(55) is an analytical method to exactly compute the SCGF for a non-Poissonian waiting time distribution of the resetting process. The calculation of the large deviation statistics of the activity for a non-Poissonian waiting time distribution $p(\tau)$ cannot, indeed, be accomplished solely on the basis of thermodynamics of quantum-jump trajectories (recalled in Subsec. 2.1) since this formalism is based on the spectral properties of a Lindbladian time-independent generator.

In Subsec. 3.2, we have derived the last renewal equation for the tilted density matrix $G(k,t)$ exploiting the renewal structure of the process, without making reference to the quantum-trajectories interpretation. In Sec. 4, we have eventually applied our general result to some simple, yet relevant, quantum optics systems, beginning with a driven two-level atom subject to incoherent decay. Here we showed that stochastic resetting can be used to tune the activity of the system from high to low values depending on the choice of the reset state $|\psi_r\rangle$ and the maximum reset time $t_{\max}$ (and the rate $\gamma$). We furthermore considered a three-level system in the shelving configuration whose quantum-jump trajectories display intermittent behavior. This intermittency can be either fostered or suppressed dependently on the reset state being chosen as the shelving state $|2\rangle$ or the inactive state $|0\rangle$, respectively.

As a future perspective it would be interesting to extend our derivation of the non-Markovian tilted generator to cases where the resetting dynamics is coupled to the underlying reset-free dynamics, similarly as in the "conditional resetting" protocol of Ref. [76]. In the

latter reference, the system is reset to a state chosen within a set of reset states conditionally on the outcome of a measurement taken right at the time of the reset. We expect the large deviation statistics of the activity to display a rich physical behavior within this conditional resetting protocol. Moreover, similar results to the ones we obtained for $\theta(k)$ in the three-level system are expected to apply to the many-body dissipative transverse-field Ising chain [12], where the onset of intermittency is controlled by the transverse field. This is true as long as the number of spins in the chain is finite and the crossover between the active and the inactive dynamical phase is smooth. In the thermodynamic limit, this crossover becomes an actual sharp dynamical first-order phase transition associated with intermittent trajectories. The analysis of such transition requires an in-depth numerical analysis beyond exact diagonalization techniques and it is therefore left as a future study.

## Acknowledgements

The authors thank M. Magoni and J. P. Garrahan for fruitful discussions.

**Funding information** G.P. acknowledges support from the Alexander von Humboldt Foundation through a Humboldt research fellowship for postdoctoral researchers. We acknowledge support from the "Wissenschaftler Rückkehrprogramm GSO/CZS" of the Carl-Zeiss-Stiftung and the German Scholars Organization e.V., from the European Union's Horizon 2020 research and innovation program under Grant Agreement No. 800942 (ErBeStA), as well as from the Baden-Württemberg Stiftung through Project No. BWST_ISF2019-23.

## A  Derivation of the non-Markovian tilted generator from the renewal equation approach

In this Appendix we report the details of the derivation of the generalized Lindblad equation (25) and of the corresponding non-Markovian tilted generator (61) using the renewal equation approach of Subsecs. 2.2 and 3.2. Since Eq. (25) is obtained upon setting $k = 0$ into Eq. (61), we directly proceed to the proof the latter.

The calculation follows along the same lines of Ref. [76] and Ref. [55], which refer to closed quantum system and classical Markovian systems, respectively, subject to stochastic resetting. We start by taking the Laplace trasform of Eq. (60) with respect to time

$$\widehat{\rho}(k,s) = \widehat{q}(s - W_k^{(0)})[|\psi_r\rangle\langle\psi_r|] + \widehat{\nu}(k,s)\widehat{q}(s - W_k^{(0)})[|\psi_r\rangle\langle\psi_r|], \tag{69}$$

where we used the shifting property of the Laplace transform and Eq. (19), with $k \neq 0$, to express the reset-free dynamics of the tilted density matrix $\rho^{(0)}(k,t)$ in terms of the reset-free tilted generator $W_k^{(0)}$. We work, for simplicity, within the assumption that the initial state coincides with the reset state $\rho(0) = |\psi_r\rangle\langle\psi_r|$. Inserting the expression $\widehat{q}(s) = (1 - \widehat{p}(s))/s$ for the Laplace transform of the survival probability into Eq. (69) one has

$$s\widehat{\rho}(k,s) - \rho(0) = W_k^{(0)}[\widehat{\rho}(k,s)] + \left(\widehat{\nu}(k,s) - \widehat{\nu}(k,s)\widehat{p}(s - W_k^{(0)}) - \widehat{p}(s - W_k^{(0)})\right)[\rho(0)]. \tag{70}$$

The term between the round brackets can be further simplified using $\widehat{p}(s) = \widehat{r}(s)\widehat{q}(s)$ in Eq. (27), which allows to write Eq. (70) as

$$s\widehat{\rho}(k,s) - \rho(0) = W_k^{(0)}[\widehat{\rho}(k,s)] + \widehat{\nu}(k,s)\rho(0) - \widehat{r}(s - W^{(0)}(k))[\widehat{\rho}(k,s)]. \tag{71}$$

Taking the inverse Laplace transform of the previous equation, exploiting the formula for the Laplace transform of the derivative on the left-hand side, one readily obtains Eq. (61) of the main text. Upon setting $k = 0$, the result of this derivation is the generalized Lindblad equation (25), as it must be.

# B  Derivation of the non-Markovian tilted generator from the quantum trajectories

In this Appendix we report the details of the derivation of Eqs. (42) and (44) with the quantum-trajectories formalism of Subsec. 3.1.

The time derivative of $\rho(N, R, t)$ in Eq. (41) has three contributions. The first one comes from differentiating the extreme of the integral over $t_R$. It therefore gives, provided $R \neq 0$, by the fundamental theorem of calculus

$$\sum_{l_1=0}^{N} \cdots \sum_{l_R=0}^{N-\sum_{i=1}^{R-1} l_i} \int_0^t dt_{R-1} \cdots \int_0^{t_2} dt_1 \mathcal{V}^{(0)}\left(N - \sum_{i=1}^{R} l_i, 0, 0\right) p(t - t_{R-1}) \mathcal{R}\mathcal{V}^{(0)}(l_R, t - t_{R-1}, 0)$$

$$\cdots p(t_1)\mathcal{R}\mathcal{V}^{(0)}(l_1, t_1, 0)[\rho(0)] =$$

$$\sum_{l_1=0}^{N} \cdots \sum_{l_{R-1}=0}^{N-\sum_{i=1}^{R-2} l_i} \int_0^t dt_{R-1} \cdots \int_0^{t_2} dt_1 p(t - t_{R-1}) P_{t-t_{R-1}}^{(0)}\left(N - \sum_{i=1}^{R-1} l_i\right)$$

$$p(t_{R-1} - t_{R-2}) P_{t_{R-1}-t_{R-2}}^{(0)}(l_{R-1}) \cdots p(t_1) P_{t_1}^{(0)}(l_1)\rho(0) = \nu(N, R, t)[\rho(0)]. \tag{72}$$

In the second equality we used that $q(t = 0) = 1$, from the definition (20), and that $\mathcal{V}^{(0)}(N, 0, 0) = \delta_{N,0}$ from Eq. (9). One then eventually recognizes the expression of $\nu(N, R, t)$ in Eq. (43). The second contribution to the time derivative of $\rho(N, R, t)$ comes from differentiating the superoperator $\mathcal{V}^{(0)}\left(N - \sum_{i=1}^{R} l_i, t - t_R, 0\right)$ on the first line of Eq. (41). This differentiation can be accomplished by exploiting the definition in Eq. (9) and the reset-free evolution equation (11) as

$$\sum_{l_1=0}^{N} \cdots \sum_{l_R=0}^{N-\sum_{i=1}^{R-1} l_i} \int_0^t dt_R \int_0^{t_R} dt_{R-1} \cdots \int_0^{t_2} dt_1 q(t - t_R) \partial_t \mathcal{V}^{(0)}\left(N - \sum_{i=1}^{R} l_i, t - t_R, 0\right)$$

$$p(t_R - t_{R-1})\mathcal{R}\mathcal{V}^{(0)}(l_R, t_R - t_{R-1}, 0) \cdots p(t_1)\mathcal{R}\mathcal{V}^{(0)}(l_1, t_1, 0)[\rho(0)] =$$

$$\mathcal{L}_0[\rho(N, R, t)] + \mathcal{L}_1[\rho(N-1, R, t)](1 - \delta_{N,0}). \tag{73}$$

The operators $\mathcal{L}_0$ and $\mathcal{L}_1$ have been defined in Eqs. (3) and (4) and they describe the reset-free dissipative time evolution. The third contribution to the time derivative of $\rho(N, R, t)$

eventually comes from differentiating $q(t - t_R)$ on the first line of Eq. (41) as

$$-\sum_{l_1=0}^{N} \cdots \sum_{l_R=0}^{N-\sum_{i=1}^{R-1} l_i} \int_0^t dt_R \int_0^{t-t_R} dt' \ldots \int_0^{t_2} dt_1 r(t') q(t - t_R - t') \mathcal{V}^{(0)}\left(N - \sum_{i=1}^{R} l_i, t - t_R, 0\right)$$

$$p(t_R - t_{R-1}) \mathcal{R} \mathcal{V}^{(0)}(l_R, t_R - t_{R-1}, 0) \ldots p(t_1) \mathcal{R} \mathcal{V}^{(0)}(l_1, t_1, 0)[\rho(0)], \quad (74)$$

where we used

$$\frac{dq(t - t_R)}{dt} = -p(t - t_R) = -\int_0^{t-t_R} dt' q(t - t_R - t') r(t'), \quad (75)$$

from the definition of the survival probability in Eq. (20) and Eq. (27) (in time domain). Equation (74) can be further simplified using the property

$$\mathcal{V}^{(0)}(N, t - t_R, 0) = \sum_{M=0}^{N} \mathcal{V}^{(0)}(M, t - t_R, t - t' - t_R) \mathcal{V}^{(0)}(N - M, t - t' - t_R, 0),$$

$$= \sum_{M=0}^{N} \mathcal{V}^{(0)}(M, t, t - t') \mathcal{V}^{(0)}(N - M, t - t' - t_R, 0). \quad (76)$$

of the reset-free evolution. The physical meaning of the previous equation is clear as it expresses $\mathcal{V}^{(0)}(N, t - t_R, 0)$ in terms of all the possible ways of distributing the $N$ jumps in $(0, t)$ between the interval $(0, t - t' - t_R)$, with $N - M$ jumps, and $(t - t' - t_R, t - t_R)$, with $M$ jumps. Equation (76) can be directly proved starting from the definition in Eq. (9). In the second line we used the fact that the superoperator $\mathcal{V}^{(0)}(M, t - t_R, t - t' - t_R) = \mathcal{V}^{(0)}(M, t, t - t')$ depends only on the difference between the final and the initial time, since the same apply for $\mathcal{S}(t, t_0)$ in Eq. (6). Exchanging the order of integration between $t_R$ and $t'$ in Eq. (71) and using Eq. (76) for $\mathcal{V}^{(0)}(N - \sum_{i=1}^{R} l_i, t - t_R, 0)$ one has

$$-\sum_{l_1=0}^{N} \cdots \sum_{l_R=0}^{N-\sum_{i=1}^{R-1} l_i} \int_0^t dt' r(t') \int_0^{t-t'} dt_R \ldots \int_0^{t_2} dt_1 q(t - t_R - t') \mathcal{V}^{(0)}\left(N - \sum_{i=1}^{R} l_i, t - t_R, 0\right)$$

$$p(t_R - t_{R-1}) \mathcal{R} \mathcal{V}^{(0)}(l_R, t_R - t_{R-1}, 0) \ldots p(t_1) \mathcal{R} \mathcal{V}^{(0)}(l_1, t_1, 0)[\rho(0)],$$

$$= -\sum_{M=0}^{N} \int_0^{t'} dt' r(t') \mathcal{V}^{(0)}(M, t, t - t')[\rho(N - M, R, t - t')]. \quad (77)$$

Putting together Eqs. (72), (73) and (77) from the derivative of $\rho(N, R, t)$ in Eq. (41) w.r.t. $t$, one obtains Eq. (42) of the main text. In order to complete the proof of the non-Markovian tilted generator in Eq. (48) one needs to write $\nu(N, R, t)$ in terms of the density matrix as in Eq. (44). The equality between Eq. (43) and (44) is proved upon decomposing $\rho(N - M, R - 1, t')$ on the right side in Eq. (44) into quantum trajectories, as in Eq. (41), and then using Eq. (76), with the intermediate steps very similar to the ones leading from Eq. (74) to (77).

The non-Markovian tilted generator in Eq. (48) is eventually attained from the discrete Laplace transform of (42) according to (47) for the reset gain term. The loss term in Eq. (77) can be similarly transformed noting the discrete convolution structure, since $\mathcal{V}^{(0)}(M, t, t - t') = \mathcal{V}^{(0)}(M, t', 0)$, and using for the discrete Laplace transform of $\mathcal{V}^{(0)}(N, t, t_0)$ in Eq. (9) the result in Eq. (18).

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
