# Peer review of "Thermodynamics of quantum-jump trajectories of open quantum systems subject to stochastic resetting"

_SciPost Physics_

## Round 2 · Referee Report · Anonymous (Referee 1) · 2022-2-2

Strengths

1- clarity

Weaknesses

1- the paper is a bit formal and technical

Report

In this paper, the authors consider Markovian open quantum systems subjected to stochastic resetting. This means that the dissipative dynamics, usually described by a Linbladian, is now
reset, randomly in time, to some specific fixed state. The study of resetting of stochastic processes has recently attracted a lot of interest, though mainly for classical systems. It was recently realized that resetting might also generate interesting behaviours of quantum dynamics. The present paper presents a study of such an open quantum systems in the presence of resetting where the times at which resetting occurs may have a non-Poissonian (i.e., non-exponential) distribution. In this case, the corresponding effective quantum dynamics is non-Markovian and it is shown to be described by a "generalized Lindblad equation". Despite of this, the authors show that it is still possible to compute the large deviation function associated to the activity (i.e., the number of quantum jumps in a given quantum trajectory) in the limit of large observation time $t$. This result relies on a renewal approach, which is a key tool to study resetting dynamics. This large deviation function is then expressed implicitly and is computed numerically for two- and three-levels systems.

The paper is quite well written and presents interesting and original results, which will may be relevant for the communities working (i) on open quantum systems and (ii) on resetting dynamics. Therefore, I would like to recommend the publication of the present paper in SciPost, which I believe is a relevant platform to publish such results. The authors may however consider the following (minor) comments:

1) The central result of the paper is the computation of the large deviation $\theta(k)$, given (in an implicit form) in Eq. (55). I think that the authors should indicate this result -- by refereeing to that equation -- when they present their main results on p. 4. This would help the reader, since the paper might appear otherwise a bit formal.

2) Above Eq. (3): "non-Hermitean" --> "non-Hermitian".

3) In Eq. (9): $d_{N-1}$ --> $dt_{N-1}$.

4) "r.h.s." below Eq. (11) -- although standard -- should be defined.

  • validity: high
  • significance: high
  • originality: good
  • clarity: top
  • formatting: excellent
  • grammar: excellent

Author:  Gabriele Perfetto  on 2022-05-30  [id 2541]

(in reply to Report 1 on 2022-02-02)

We thank the Referee for the supportive report and the careful reading of our manuscript. In the following, we address their report.

The Referee writes:

Therefore, I would like to recommend the publication of the present paper in SciPost, which I believe is a relevant platform to publish such results. The authors may however consider the following (minor) comments: 1) The central result of the paper is the computation of the large deviation $\theta(k)$, given (in an implicit form) in Eq.~(55). I think that the authors should indicate this result -- by refereeing to that equation -- when they present their main results on p. 4. This would help the reader, since the paper might appear otherwise a bit formal.

Our response:

We thank the Referee for this suggestion, which helped in improving the presentation of the main result of the paper. We have explicitly referenced this result and Eqs.~(50)-(55) for the calculation of $\theta(k)$ in the introduction on page 4 of the revised manuscript. We have further referenced Eqs.~(50)-(55) at the beginning of Sec.~3 on page 14 of the revised manuscript. The sentences we added are highlighted in blue.

The Referee writes

2) Above Eq.~(3): "non-Hermitean" $\to$ "non-Hermitian". 3) In Eq.~(9): $\mbox{d}N-1 \to \mbox{dt}_{N-1}$. 4) "r.h.s." below Eq. (11) -- although standard -- should be defined.

Our response

We thank the Referee for pointing out these issues and typos in our manuscript. We have corrected all the points 2)-4) mentioned by the Referee.

Anonymous on 2022-06-15  [id 2579]

(in reply to Gabriele Perfetto on 2022-05-30 [id 2541])

In this revised version, the authors have satisfactorily taken into account my comments. Therefore, I wish to recommend the publication of the present paper in SciPost.

---

## Round 2 · Referee Report · Krishnendu Sengupta (Referee 2) · 2022-2-7

Report

The paper discusses a theoretical Lindblad operator based approach to the problem of stochastic resetting. The paper provides the detailed derivation of a Lindblad based formalism for generic reset problem. Their analysis constitutes a generalization of the Poissonian resetting problem which is reproduced as a special case. However, the illustrative examples provided in the paper are for somewhat simple systems ( two and three level optics problems). A much more non-trivial example which, in my opinion, could be worked out within their approach would be the reset dynamics of 1D Ising model in a transverse field. Furthermore, the analysis section of the paper is a bit dense; it would help an average reader if it can be re-written in a slightly more pedagogical manner. For example, the authors do not quite explain how they get Eqs 5 and 6 from Eq 4. I did look up the refs they provide, and it may take quite a bit of time for the reader to figure this out. So I would suggest a bit of explanatory sentences/formulae connecting these equations even if they have been done before (may be put in the appendix, if the authors feel that it would disturb the flow).

Requested changes

  1. Include a slightly more complicated example to demonstrate the power of the method.

  2. Make the presentation slightly more accessible.

  • validity: good
  • significance: good
  • originality: high
  • clarity: ok
  • formatting: perfect
  • grammar: good

Author:  Gabriele Perfetto  on 2022-05-30  [id 2542]

(in reply to Report 2 by Krishnendu Sengupta on 2022-02-07)

We thank the Referee for the careful reading of our manuscript. In the following, we address their report. We have highlighted in blue the most important changes we did in the text while revising it.

The referee writes:

However, the illustrative examples provided in the paper are for somewhat simple systems ( two and three level optics problems). A much more non-trivial example which, in my opinion, could be worked out within their approach would be the reset dynamics of 1D Ising model in a transverse field. [...] 1. Include a slightly more complicated example to demonstrate the power of the method.

Our response:

The main result of the manuscript is the establishment of the analytical result, summarized in Eqs.~(50)-(55), as a general method to exactly compute the large-deviation statistics in open quantum systems subject to non-Poissonian resetting. This method is general, as we discuss in the manuscript, as it can be applied to any open quantum system whose evolution in the absence of resetting is ruled by the Lindblad equation. This result is important as for non-Poissonian resetting the SCGF cannot be computed by other means, such as the thermodynamics of quantum trajectories formalism of Ref.~[9], since the tilted generator is time dependent (in a non local way in its most natural representation). In addition, we mention, as written also in the reply to the Referee 3, that the characterization of the SCGF in the context of non-Poissonian resetting is a tough and demanding problem even at the numerical level. As a matter of fact, the SCGF $\theta(k)$ encodes the information about the rare tails of the activity distribution in the long time limit $t \to \infty$. These tails are exponentially suppressed as a function of $t$ and therefore a larger and larger numerical sampling of quantum trajectories is needed as $t \to \infty$. It is also interesting to mention that for Markovian quantum systems, trajectories at the tails of the activity distribution can be efficiently simulated by resorting to the quantum Doob tranformation of Ref.~[9,17], which allows to construct a Markovian Lindblad generator which presents as typical trajectories the rare trajectories of the original Lindblad dynamics. In the present context of the generalized non-Markovian Lindblad equation (25), it is not, however, known how to construct the Doob transformation.

The availability of the analytical result in Eqs.~(50)-(55) therefore provides a remarkable prediction for the SCGF, which could not be computed analytically otherwise, and that would require, to the best of our knowledge, a demanding and inefficient sampling for the numerical estimation.

It is fundamental to stress that the aforementioned difficulties apply already to the class of systems we study in Sec.~4, the two and three-level systems. The latter systems are therefore valuable examples which show "the power of the method'', using the words of the Referee, since their quantum jump statistics cannot be computed for non-Poissonian resetting without the scheme we present. The "power of the method'', in this perspective, relies in capability of providing analytical predictions for the highly challenging task of characterizing the long time rare-event statistics of the activity obtained from a non-Markovian dynamics. We have improved the presentation of the manuscript by adding a discussion of these points in the Introduction, in Sec.~2,3, at the beginning of Sec.~4 on page 19, and in the conclusion.

The Referee also suggests considering the one-dimensional dissipative transverse-field Ising model as an additional example. We thank the Referee for this suggestion, which allows us to draw a nice connection with the physics of the three-level system discussed in the text. The quantum-jump statistics of the Ising chain, without stochastic resetting, has been characterized in Ref.~[12] of the manuscript. In this work, it has been shown that the transverse field drives the system across a first-order (discontinuous) phase transition, where quantum trajectories are active for high values of the transverse field, and inactive for low values of this parameter. For intermediate values, intermittency in the quantum trajectories is observed. This first-order transition has been predicted in the thermodynamic limit on the basis of mean-field calculations. At the numerical level, via exact diagonalization techniques for $N=7$ spins, the transition clearly turns into a smoothened crossover around $k=0$ in the SCGF $\theta(k)$. This smoothened crossover is analogous to the one we observe in Subsec.~4.2 for the three-level system. Also in this case, as a matter of fact, quantum-jump trajectories are intermittent, as discussed in the text.

We therefore expect, for a finite number $N$ of spins, for the dissipative transverse-field Ising chain similar results to the ones we present for the three-level system. The results for the three-level system are therefore also paradigmatic for the effect of resetting on possibly many-body quantum systems, with a finite number $N$ of particles, displaying intermittency in the quantum-jump trajectories. In the thermodynamic limit $N \to \infty$, where in Ising chain the crossover becomes an actual discontinuous transition, differences from the three-level intermittent physics are clearly expected. The quantification of this aspect requires a large-scale numerical study on its own which goes beyond the scope of the manuscript as one needs to perform an advanced numerical analysis beyond exact diagonalization. We have discussed these aspects in Subsec.~4.2 and in the conclusions. The corresponding parts of the text are highlighted in blue.

The referee writes:

Furthermore, the analysis section of the paper is a bit dense; it would help an average reader if it can be re-written in a slightly more pedagogical manner. For example, the authors do not quite explain how they get Eqs 5 and 6 from Eq 4. I did look up the refs they provide, and it may take quite a bit of time for the reader to figure this out. So I would suggest a bit of explanatory sentences/formulae connecting these equations even if they have been done before (may be put in the appendix, if the authors feel that it would disturb the flow). [...] 2. Make the presentation slightly more accessible.

Our response:

We thank the Referee for pointing out this issue. The explanation of the steps from Eqs.~(4) to (5) and (6) was, indeed, not clear. We have added an extended discussion of the interpretation of Eq.~(5). This part of the text is right after Eq.~(6) and it is highlighted in blue. We have also therein cited Ref.~[8], which was not cited in the first version of the manuscript at that point. Reference [8], is the first, to our knowledge, work where the decomposition in Eq.~(5) of the Lindblad time evolution operator is reported. The detailed derivation of Eq.~(5) of the manuscript can be found in the second and third pages of Ref.~[8] (Subsec.~II A ``Two-level system''), with Eq.~(20) therein corresponding to Eq.~(5) of the manuscript.

---

## Round 2 · Referee Report · Anonymous (Referee 3) · 2022-2-18

Strengths

1- Very clear, especially for experts 2-Potentially very useful approach for simulating non-Markovian dynamics

Weaknesses

1-Most of the results are formal 2- Examples seem a bit too limited 3- Motivation for stochastic reset not clear

Report

The authors show how the stochastic resetting dynamics of [39,40] can be used to simulate non-Markovian quantum dynamics. They likewise derive formal results for the quantum jumps statistics. They apply their method to two examples, a two and three-level atom system.

I think that the paper is interesting and represents an important contribution to the theory of open quantum systems. I especially like the part of simulating non-Markovian dynamics.

Requested changes

1- The authors should explain better why their results on the quantum jump statistics are not merely formal and can be used to solve difficult problems. 2- In the introduction the authors introduce stochastic resets with a references. It would be nice for the reader to have more on the motivation for this process: for example, why stochastic, instead of deterministic reset? Why reset at all? Is this something that gives interesting physics and can be simulated or does it model something realistic? 3- Can the authors give a bit more elaborate example than a 3-level system? Or explain why the problem cannot be treated with more elementary approaches

  • validity: top
  • significance: good
  • originality: high
  • clarity: top
  • formatting: excellent
  • grammar: perfect

Author:  Gabriele Perfetto  on 2022-05-30  [id 2543]

(in reply to Report 3 on 2022-02-18)

We thank the Referee for the careful reading of our manuscript and for pointing out some fundamental aspects which were not clearly explained in the text. In the following, we address their report. We have highlighted in blue the most important changes we did in the text while revising it.

The referee writes:

1-The authors should explain better why their results on the quantum jump statistics are not merely formal and can be used to solve difficult problems.

Our response: We thank the Referee for pointing out this problem. We believe that the judgment about our results being "merely formal'' was motivated by a lack of clarity of the presentation, as we now explain. "Formal results'', within our understanding, are restatements of already known facts, which do not lead to new scientific predictions. Our results, on the contrary, posses a predictive power as they represent a generically applicable method for the exact calculation of the SCGF associated to the activity statistics for a non-Markovian quantum dynamics. This is a ``difficult problem'', which cannot be solved by other means, for the following reasons.

First, there are, in general, no analytical methods to accomplish such a task. The most used analytical method, indeed, to compute the SCGF of the activity statistics in open quantum systems is the thermodynamics of quantum-jump trajectories formalism of Ref.~[9]. This method relies on the spectral properties of the tilted $\textit{time-independent}$ Lindbladian generator controlling the rare trajectories statistics of a Markovian-Lindblad dynamics. In the case of non-Poissonian resetting, which is the main focus of the paper, the tilted generator is $\textit{time-dependent and non-local in time}$ in its most natural representation in Eq.~(48). The thermodynamics of quantum-jump trajectories method cannot be therefore applied in this case. It is therefore not clear whether it is analytically possible at all to compute the SCGF. The scheme we develop in the manuscript to analytically compute the SCFG, summarized in Eqs.~(50)-(55), is to our knowledge the only analytical method to compute the SCGF for the non-Markovian generator (48) describing open quantum systems subject to non-Poissonian resetting.

Second, from the numerical perspective, the problem of computing the SCGF for non-Poissonian resetting is very challenging. As a matter of fact, the SCGF $\theta(k)$ encodes the information about the rare tails of the activity distribution in the long time limit $t \to \infty$. These tails are exponentially suppressed as a function of $t$ and therefore a larger and larger numerical sampling of quantum trajectories is needed as $t \to \infty$. It is also interesting to mention that for Markovian quantum systems, trajectories at the tails of the activity distribution can be efficiently simulated by resorting to the quantum Doob tranformation of Ref.~[9,17], which allows to construct a Markovian Lindblad generator which presents as typical trajectories the rare trajectories of the original Lindblad dynamics. In the present context of the generalized non-Markovian Lindblad equation (25), it is not, however, known how to construct the Doob transformation.

In this sense the presence of the analytical result in Eqs.~(50)-(55) is a step-change as it provides an analytical prediction for the SCGF for a non-Markovian quantum dynamics, which could not be analytically obtained by other means and that would require a challenging and inefficient numerical analysis.

We have improved the presentation of our results by discussing in a more detailed way the aforementioned points in the Introduction, in Sec.~2 and 3 and in the conclusions. The revised parts of the manuscript are highlighted in blue.

The referee writes:

2-In the introduction the authors introduce stochastic resets with a references. It would be nice for the reader to have more on the motivation for this process: for example, why stochastic, instead of deterministic reset? Why reset at all? Is this something that gives interesting physics and can be simulated or does it model something realistic?

Our response: We thank the Referee for pointing out this point. We, indeed, realized that stochastic resetting was not clearly introduced in the first version of the manuscript. We have added a paragraph in the introduction where stochastic resetting is motivated in order to model many different physical phenomena such as animal foraging for food, biophysics models for proteins binding and searching algorithms. We have then included Refs.~[39-52]. We further mentioned that resetting is also widely studied in physics as a paradigmatic dynamics attaining a non-equilibrium stationary state and, possibly, dynamical transitions in the relaxation to such stationary state. We have accordingly added Ref.~[56]. The revised paragraph of the introduction is highlighted in blue.

The referee writes:

3-Can the authors give a bit more elaborate example than a 3-level system? Or explain why the problem cannot be treated with more elementary approaches

Our response: We find this point raised by the Referee very much related to the point 1 discussed above and therefore that it is caused by a lack of clarity in the presentation.

First of all, in the case the Referee is aware of "more elementary approaches'' to compute the SCGF in the framework analyzed in the manuscript, we would be very much interested in a pointer towards the relevant literature. To our knowledge, the main "elementary approach'', to use the words of the Referee, to compute the large-deviation statistics of quantum-jumps in Markovian-Lindblad open quantum systems is the thermodynamics of quantum-jumps (Ref.~[9] of the manuscript). In the case addressed in the manuscript of non-Poissonian resetting, this method cannot be applied because it relies on the spectrum of a time-independent tilted generator, while the tilted generator we derive for non-Poissonian resetting depends on time (in a non-local way in its most natural representation).

The main result of the manuscript is the establishment of the analytical result, summarized in Eqs.~(50)-(55), as a general method to exactly compute the large-deviation statistics in open quantum systems subject to non-Poissonian resetting. In Sec.~4, we have chosen the two and three-level systems just to exemplify our generic findings. We, however, emphasize that already for the two and three-level systems with non-Poissonian resetting there are to the best of our knowledge no other analytical methods (see the discussion above) to compute exactly the SCGF. Numerical methods in this context are as well very limited, as explained in the reply to point 1.

---

## Editorial Decision

resubmitted